# Towards Principled Graph Transformers

**Luis Müller**
RWTH Aachen University
luis.mueller@cs.rwth-aachen.de

**Daniel Kusuma**
RWTH Aachen University

**Blai Bonet**
Universitat Pompeu Fabra

**Christopher Morris**
RWTH Aachen University

## Abstract

The expressive power of graph learning architectures based on the $k$-dimensional Weisfeiler–Leman ($k$-WL) hierarchy is well understood. However, such architectures often fail to deliver solid predictive performance on real-world tasks, limiting their practical impact. In contrast, global attention-based models such as graph transformers demonstrate strong performance in practice. However, comparing their expressivity with the $k$-WL hierarchy remains challenging, particularly since attention-based architectures rely on positional or structural encodings. To address this, we show that the recently proposed Edge Transformer, a global attention model operating on node pairs instead of nodes, has 3-WL expressive power when provided with the right tokenization. Empirically, we demonstrate that the Edge Transformer surpasses other theoretically aligned architectures regarding predictive performance and is competitive with state-of-the-art models on algorithmic reasoning and molecular regression tasks while not relying on positional or structural encodings. Our code is available at https://github.com/luis-mueller/towards-principled-gts.

## 1 Introduction

Graph Neural Networks (GNNs) are the de-facto standard in graph learning [17, 44, 29, 51] but suffer from limited expressivity in distinguishing non-isomorphic graphs in terms of the 1-*dimensional Weisfeiler–Leman algorithm* (1-WL) [36, 51]. Hence, recent works introduced *higher-order* GNNs, aligned with the $k$-dimensional Weisfeiler–Leman ($k$-WL) hierarchy for graph isomorphism testing [1, 34, 36, 37, 39], resulting in more expressivity with an increase in $k > 1$. The $k$-WL hierarchy draws from a rich history in graph theory and logic [3, 4, 5, 10, 50], offering a deep theoretical understanding of $k$-WL-aligned GNNs. While theoretically intriguing, higher-order GNNs often fail to deliver state-of-the-art performance on real-world problems, making theoretically grounded models less relevant in practice [1, 37, 39]. In contrast, graph transformers [18, 20, 32, 42, 53] recently demonstrated state-of-the-art empirical performance. However, they draw their expressive power mostly from positional/structural encodings (PEs), making it difficult to understand these models in terms of an expressivity hierarchy such as the $k$-WL. While a few works theoretically aligned graph transformers with the $k$-WL hierarchy [27, 28, 54], we are not aware of any works reporting empirical results for 3-WL-equivalent graph transformers on established graph learning datasets.

In this work, we aim to set the ground for graph learning architectures that are theoretically aligned with the higher-order Weisfeiler–Leman hierarchy while delivering strong empirical performance and, at the same time, demonstrate that such an alignment creates powerful synergies between transformers and graph learning. Hence, we close the gap between theoretical expressivity and real-world predictive power. To this end, we apply the *Edge Transformer* (ET) architecture, initially

38th Conference on Neural Information Processing Systems (NeurIPS 2024).

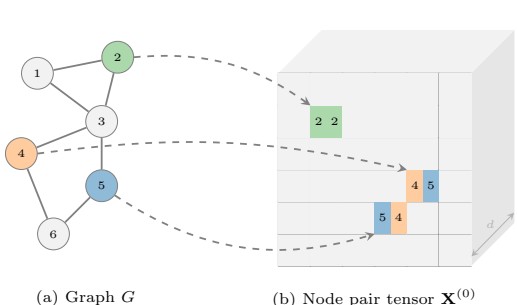

(a) Graph $G$      (b) Node pair tensor $\mathbf{X}^{(0)}$

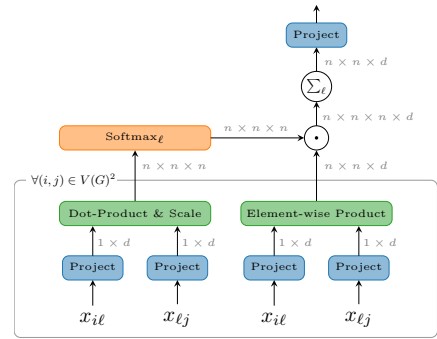

Figure 1: Tokenization of the Edge Transformer. Given a graph $G$, we construct a 3D tensor where we embed information from each node pair into a $d$ dimensional vector.

Figure 2: Tensor operations in a single triangular attention head; see Algorithm 1 for a comparison to standard attention in pseudo-code.

developed for *systematic generalization* problems [6], to the field of graph learning. Systematic (or compositional) generalization refers to the ability of a model to generalize to complex novel concepts by combining primitive concepts observed during training, posing a challenge to even the most advanced models such as GPT-4 [15].

Specifically, we contribute the following:

1. We propose a concrete implementation of the Edge Transformer that readily applies to various graph learning tasks.
2. We show theoretically that this Edge Transformer implementation is as expressive as the 3-WL *without* the need for positional/structural encodings.
3. We demonstrate the benefits of aligning models with the $k$-WL hierarchy by leveraging well-established results from graph theory and logic to develop a theoretical understanding of systematic generalization in terms of first-order logic statements.
4. We demonstrate the superior empirical performance of the resulting architecture compared to a variety of other theoretically motivated models, as well as competitive performance compared to state-of-the-art models in molecular regression and neural algorithmic reasoning tasks.

## 2 Related work

Many graph learning models with higher-order WL expressive power exist, notably $\delta$-$k$-GNNs [37], SpeqNets [39], $k$-IGNs [35, 34], PPGN [33], and the more recent PPGN++ [41]. Moreover, Lipman et al. [31] devise a low-rank attention module possessing the same power as the folklore 2-WL and Bodnar et al. [8] propose CIN with an expressive power of at least 3-WL. For an overview of Weisfeiler–Leman in graph learning, see Morris et al. [38].

Many graph transformers exist, notably Graphormer [53] and GraphGPS [42]. However, state-of-the-art graph transformers typically rely on positional/structural encodings, which makes it challenging to derive a theoretical understanding of their expressive power. The Relational Transformer (RT) [12] operates over both nodes and edges and, similar to the ET, builds relational representations, that is, representations on edges. Although the RT integrates edge information into self-attention and hence does not need to resort to positional/structural encodings, the RT is theoretically poorly understood, much like other graph transformers. Graph transformers with higher-order expressive power are Graphormer-GD [54] and TokenGT [28] as well as the higher-order graph transformers in Kim et al. [27]. However, Graphormer-GD is strictly less expressive than the 3-WL [54]. Further, Kim et al. [27] and Kim et al. [28] align transformers with $k$-IGNs and, thus, obtain the theoretical expressive power of the corresponding $k$-WL but do not empirically evaluate their transformers for $k > 2$. In addition, these higher-order transformers suffer from a $\mathcal{O}(n^{2k})$ runtime and memory complexity. For $k = 3$,

the ET offers provable 3-WL expressivity with $\mathcal{O}(n^3)$ runtime and memory complexity, several orders of magnitude more efficient than the corresponding 3-WL expressive transformer in Kim et al. [28]. For an overview of graph transformers, see Müller et al. [40].

Finally, systematic generalization has recently been investigated both empirically and theoretically [6, 15, 26, 43]. In particular, Dziri et al. [15] demonstrate that compositional generalization is lacking in state-of-the-art transformers such as GPT-4.

## 3    Edge Transformers

The ET was originally designed to improve the systematic generalization abilities of machine learning models. To borrow the example from Bergen et al. [6], a model that is presented with relations such as MOTHER$(x, y)$, indicating that $y$ is the mother of $x$, could generalize to a more complex relation GRANDMOTHER$(x, z)$, indicating that $z$ is the grandmother of $x$ if MOTHER$(x, y) \wedge$ MOTHER$(y, z)$ holds. The particular form of attention used by the ET, which we will formally introduce hereafter, is designed to explicitly model such more complex relations. Indeed, leveraging our theoretical results of Section 4, in Section 5, we formally justify the ET for performing systematic generalization. We will now formally define the ET; see Appendix D for a complete description of our notation.

In general, the ET operates on a graph $G$ with nodes $V(G)$ and consecutively updates a 3D tensor state $\boldsymbol{X} \in \mathbb{R}^{n \times n \times d}$, where $d$ is the embedding dimension and $\boldsymbol{X}_{ij}$ or $\boldsymbol{X}(\boldsymbol{u})$ denotes the representation of the node pair $\boldsymbol{u} := (i, j) \in V(G)^2$; see Figure 1 for a visualization of this construction. Concretely, the $t$-th ET layer computes

$$\boldsymbol{X}_{ij}^{(t)} := \mathsf{FFN}\big(\boldsymbol{X}_{ij}^{(t-1)} + \mathsf{TriAttention}\big(\mathsf{LN}\big(\boldsymbol{X}_{ij}^{(t-1)}\big)\big)\big),$$

for each node pair $(i, j)$, where FFN is a feed-forward neural network, LN denotes layer normalization [2] and TriAttention is defined as

$$\mathsf{TriAttention}(\boldsymbol{X}_{ij}) := \sum_{l=1}^{n} \alpha_{ilj} \boldsymbol{V}_{ilj}, \tag{1}$$

which computes a tensor product between a three-dimensional *attention tensor* $\alpha$ and a three-dimensional *value tensor* $\mathbf{V}$, by multiplying and summing over the second dimension. Here,

$$\alpha_{ilj} := \underset{l \in [n]}{\mathsf{softmax}}\Big(\frac{1}{\sqrt{d}} \boldsymbol{X}_{il} \boldsymbol{W}^Q \big(\boldsymbol{X}_{lj} \boldsymbol{W}^K\big)^T\Big) \in \mathbb{R} \tag{2}$$

is the attention score between the features of tuples $(i, l)$ and $(l, j)$, and

$$\boldsymbol{V}_{ilj} := \boldsymbol{X}_{il} \boldsymbol{W}^{V_1} \odot \boldsymbol{X}_{lj} \boldsymbol{W}^{V_2}, \tag{3}$$

we call *value fusion* of the tuples $(i, l)$ and $(l, j)$ with $\odot$ denoting element-wise multiplication. Moreover, $\boldsymbol{W}^Q, \boldsymbol{W}^K, \boldsymbol{W}^{V_1}, \boldsymbol{W}^{V_2} \in \mathbb{R}^{d \times d}$ are learnable projection matrices; see Figure 2 for an overview of the tensor operations in triangular attention and see Algorithm 1 for a comparison to standard attention [46] in pseudo-code. Note that similar to standard attention, triangular attention can be straightforwardly extended to multiple heads.

As we will show in Section 4, the ET owes its expressive power to the special form of triangular attention. In our implementation of the ET, we use the following tokenization, which is sufficient to obtain our theoretical results.

**Tokenization**   We consider graphs $G := (V(G), E(G), \ell)$ with $n$ nodes and without self-loops, where $V(G)$ is the set of nodes, $E(G)$ is the set of edges, and $\ell : V(G) \to \mathbb{N}$ assigns an initial *color* to each node. We construct a feature matrix $\boldsymbol{F} \in \mathbb{R}^{n \times p}$ that is *consistent* with $\ell$, i.e., for nodes $i$ and $j$ in $V(G)$, $\boldsymbol{F}_i = \boldsymbol{F}_j$ if and only if, $\ell(i) = \ell(j)$. Note that, for a finite subset of $\mathbb{N}$, we can always construct $\boldsymbol{F}$, e.g., using a one-hot encoding of the initial colors. Additionally, we consider an edge feature tensor $\boldsymbol{E} \in \mathbb{R}^{n \times n \times q}$, where $\boldsymbol{E}_{ij}$ denotes the edge feature of the edge $(i, j) \in E(G)$. If no edge features are available, we randomly initialize learnable vectors $\boldsymbol{x}_1, \boldsymbol{x}_2 \in \mathbb{R}^q$ and assign $\boldsymbol{x}_1$ to

**Algorithm 1** Comparison between standard attention and triangular attention in PYTORCH-like pseudo-code.

| | |
|---|---|
| **function** ATTENTION($\mathbf{X} : n \times d$) | **function** TRI_ATTENTION($\mathbf{X} : n \times n \times d$) |
| $\quad \mathbf{Q}, \mathbf{K}, \mathbf{V} \leftarrow$ linear($\mathbf{X}$).chunk(3) | $\quad \mathbf{Q}, \mathbf{K}, \mathbf{V}^1, \mathbf{V}^2 \leftarrow$ linear($\mathbf{X}$).chunk(4) |
| $\quad$ # no op | $\quad \mathbf{V} \leftarrow$ einsum($ild, ljd \rightarrow iljd, \mathbf{V}^1, \mathbf{V}^2$) |
| $\quad \tilde{\mathbf{A}} \leftarrow$ einsum($id, jd \rightarrow ij, \mathbf{Q}, \mathbf{K}$) | $\quad \tilde{\mathbf{A}} \leftarrow$ einsum($ild, ljd \rightarrow ilj, \mathbf{Q}, \mathbf{K}$) |
| $\quad \mathbf{A} \leftarrow$ softmax($\tilde{\mathbf{A}}/\sqrt{d}, -1$) | $\quad \mathbf{A} \leftarrow$ softmax($\tilde{\mathbf{A}}/\sqrt{d}, -2$) |
| $\quad \mathbf{O} \leftarrow$ einsum($ij, jd \rightarrow id, \mathbf{A}, \mathbf{V}$) | $\quad \mathbf{O} \leftarrow$ einsum($ilj, iljd \rightarrow ijd, \mathbf{A}, \mathbf{V}$) |
| $\quad$ **return** linear($\mathbf{O}$) | $\quad$ **return** linear($\mathbf{O}$) |
| **end function** | **end function** |

$\boldsymbol{E}_{ij}$ if $(i, j) \in E(G)$. Further, for all $i \in V(G)$, we assign $\boldsymbol{x}_2$ to $\boldsymbol{E}_{ii}$. Lastly, if $(i, j) \notin E(G)$ and $i \neq j$, we set $\boldsymbol{E}_{ij} = \mathbf{0}$. We then construct a 3D tensor of input tokens $\boldsymbol{X} \in \mathbb{R}^{n \times n \times d}$, such that for node pair $(i, j) \in V(G)^2$,

$$\boldsymbol{X}_{ij} \coloneqq \phi\big([\boldsymbol{E}_{ij} \quad \boldsymbol{F}_i \quad \boldsymbol{F}_j]\big), \tag{4}$$

where $\phi \colon \mathbb{R}^{2p+q} \to \mathbb{R}^d$ is a neural network. Extending Bergen et al. [6], our tokenization additionally considers node features, making it more appropriate for the graph learning setting.

**Efficiency** The triangular attention above imposes a $\mathcal{O}(n^3)$ runtime and memory complexity, which is significantly more efficient than other transformers with 3-WL expressive power, such as the higher-order transformers in Kim et al. [27] and Kim et al. [28] with a runtime of $\mathcal{O}(n^6)$. Nonetheless, the ET is still significantly less efficient than most graph transformers, with a runtime of $\mathcal{O}(n^2)$ [32, 42, 53]. Thus, the ET is currently only applicable to mid-sized graphs; see Section 7 for an extended discussion of this limitation.

**Positional/structural encodings** Additionally, GNNs and graph transformers often benefit empirically from added positional/structural encodings [13, 32, 42]. We can easily add PEs to the above tokens with the ET. Specifically, we can encode any PEs for node $i \in V(G)$ as an edge feature in $\boldsymbol{E}_{ii}$ and any PEs between a node pair $(i, j) \in V(G)^2$ as an edge feature in $\boldsymbol{E}_{ij}$. Note that typically, PEs between pairs of nodes are incorporated during the attention computation of graph transformers [32, 53]. However, in Section 6, we demonstrate that simply adding these PEs to our tokens is also viable for improving the empirical results of the ET.

**Readout** Since the Edge Transformer already builds representations on node pairs, making predictions for node pair- or edge-level tasks is straightforward. Specifically, let $L$ denote the number of Edge Transformer layers. Then, for a node pair $(i, j) \in V(G)^2$, we simply readout $\boldsymbol{X}_{ij}^{(L)}$, where on the edge-level we restrict ourselves to the case where $(i, j) \in E(G)$. In the case of nodes, we can for example read out the diagonal of $\boldsymbol{X}^{(L)}$, that is, the representation for node $i \in V(G)$ is $\boldsymbol{X}_{ii}^{(L)}$. In addition to the diagonal readout, we also design a more sophisticated read out strategy for nodes which we describe in Appendix A.1.

With tokenization and readout as defined above, the ET can now be used on many graph learning problems, encoding both node and edge features and making predictions for node pair-, edge-, node-, and graph-level tasks. We refer to a concrete set of parameters of the ET, including tokenization and positional/structural encodings, as a *parameterization*. We now move on to our theoretical result, showing that the ET has the same expressive power as the 3-WL.

## 4 The expressivity of Edge Transformers

Here, we relate the ET to the *folklore* Weisfeiler–Leman ($k$-FWL) hierarchy, a variant of the $k$-WL hierarchy for which, for $k > 2$, $(k-1)$-FWL is as expressive as $k$-WL [19]. Specifically, we show that the ET can simulate the 2-FWL, resulting in 3-WL expressive power. To this end, we briefly introduce

the 2-FWL and then show our result. For detailed background on the $k$-WL and $k$-FWL hierarchy, see Appendix D.

**Folklore Weisfeiler–Leman**   Let $G \coloneqq (V(G), E(G), \ell)$ be a node-labeled graph. The 2-FWL colors the tuples from $V(G)^2$, similar to the way the 1-WL colors nodes [36]. In each iteration, $t \geq 0$, the algorithm computes a *coloring* $C_t^{2,\mathrm{F}} \colon V(G)^2 \to \mathbb{N}$ and we write $C_t^{2,\mathrm{F}}(i,j)$ or $C_t^{2,\mathrm{F}}(\boldsymbol{u})$ to denote the color of tuple $\boldsymbol{u} \coloneqq (i,j) \in V(G)^2$ at iteration $t$. For $t = 0$, we assign colors to distinguish pairs $(i,j)$ in $V(G)^2$ based on the initial colors $\ell(i), \ell(j)$ of their nodes and whether $(i,j) \in E(G)$ or $i = j$. For a formal definition of the initial node pair colors, see Appendix D. Then, for each iteration, $t > 0$, the coloring $C_t^{2,\mathrm{F}}$ is defined as

$$C_t^{2,\mathrm{F}}(i,j) \coloneqq \mathsf{recolor}\big((C_{t-1}^{2,\mathrm{F}}(i,j),\, M_{t-1}(i,j))\big),$$

where recolor injectively maps the above pair to a unique natural number that has not been used in previous iterations and

$$M_{t-1}(i,j) \coloneqq \{\!\{(C_{t-1}^{2,\mathrm{F}}(i,l),\, C_{t-1}^{2,\mathrm{F}}(l,j)) \mid l \in V(G)\}\!\}.$$

We show that the ET can simulate the 2-FWL, resulting in at least 3-WL expressive power. Further, we show that the ET is also, at most, as expressive as the 3-WL. As a result, we obtain the following theorem; see Appendix E for a formal statement and proof details.

**Theorem 1** (Informal). *The ET has exactly* 3-*WL expressive power.*

Note that following previous works [33, 37, 39], our expressivity result is *non-uniform* in that our result only holds for an arbitrary but fixed graph size $n$; see Proposition 7 and Proposition 8 for the complete formal statements and proof of Theorem 1.

In the following, we provide some intuition of how the ET can simulate the 2-FWL. Given a tuple $(i,j) \in V(G)^2$, we encode its color at iteration $t$ with $\boldsymbol{X}_{ij}^{(t)}$. Further, to represent the multiset

$$\{\!\{(C_{t-1}^{2,\mathrm{F}}(i,l),\, C_{t-1}^{2,\mathrm{F}}(l,j)) \mid l \in V(G)\}\!\},$$

we show that it is possible to encode the pair of colors

$$(C_{t-1}^{2,\mathrm{F}}(i,l),\, C_{t-1}^{2,\mathrm{F}}(l,j)) \quad \text{via} \quad \boldsymbol{X}_{il}^{(t-1)} \boldsymbol{W}^{V_1} \odot \boldsymbol{X}_{lj}^{(t-1)} \boldsymbol{W}^{V_2},$$

for node $l \in V(G)$. Finally, triangular attention in Equation (1), performs weighted sum aggregation over the 2-tuple of colors $(C_{t-1}^{2,\mathrm{F}}(i,l),\, C_{t-1}^{2,\mathrm{F}}(l,j))$ for each $l$, which we show is sufficient to represent the multiset; see Appendix E. For the other direction, namely that the ET has at most 3-WL expressive power, we simply show that the recolor function can simulate the value fusion in Equation (3), as well as the triangular attention in Equation (1).

Interestingly, our proofs do not resort to positional/structural encodings. The ET draws its 3-WL expressive power from its aggregation scheme, the triangular attention. In Section 6, we demonstrate that this also holds in practice, where the ET performs strongly without additional encodings. In what follows, we use the above results to derive a more principled understanding of the ET in terms of systematic generalization, for which it was originally designed. Thereby, we demonstrate that graph theoretic results can also be leveraged in other areas of machine learning, further highlighting the benefits of theoretically grounded models.

## 5   The logic of Edge Transformers

After borrowing the ET from systematic generalization in the previous section, we now return the favor. Specifically, we use a well-known connection between graph isomorphism and first-order logic to obtain a theoretical justification for systematic generalization reasoning using the ET. Recalling the example around the GRANDMOTHER relation composed from the more primitive MOTHER relation in Section 3, Bergen et al. [6] go ahead and argue that since self-attention of standard transformers is defined between pairs of nodes, learning explicit representations of GRANDMOTHER is impossible and that learning such representations implicitly incurs a high burden on the learner. Conversely, the

authors argue that since the ET computes triangular attention over triplets of nodes and computes explicit representations between node pairs, the Edge Transformer can systematically generalize to relations such as GRANDMOTHER. While Bergen et al. [6] argue the above intuitively, we will now utilize the connection between first-order logic (FO-logic) and graph isomorphism established in Cai et al. [10] to develop a theoretical understanding of systematic generalization; see Appendix D for an introduction to first-order logic over graphs. We will now briefly introduce the most important concepts in Cai et al. [10] and then relate them to systematic generalization of the ET and similar models.

**Language and configurations** Here, we consider FO-logic statements with counting quantifiers and denote with $\mathcal{C}_{k,m}$ the language of all such statements with at most $k$ variables and quantifier depth $m$. A *configuration* is a map between first-order variables and nodes in a graph. Concretely, configurations let us define a statement $\varphi$ in first-order logic, such as three nodes forming a triangle, without speaking about concrete nodes in a graph $G = (V(G), E(G))$. Instead, we can use a configuration to map the three variables in $\varphi$ to nodes $v, w, u \in V(G)$ and evaluate $\varphi$ to determine whether $v, w$ and $u$ form a triangle in $G$. Of particular importance to us are $k$-configurations $f$ where we map $k$ variables $x_1, \ldots, x_k$ in a FO-logic statement to a $k$-tuple $\boldsymbol{u} \in V(G)^k$ such that $\boldsymbol{u} = (f(x_1), \ldots, f(x_k))$. This lets us now state the following result in Cai et al. [10], relating FO-logic satisfiability to the $k$-FWL hierarchy. Here, $C_t^{k,\mathrm{F}}$ denotes the coloring of the $k$-FWL after $t$ iterations; see Appendix D for a precise definition.

**Theorem 2** (Theorem 5.2 [10], informally). *Let $G := (V(G), E(G))$ and $H := (V(H), E(H))$ be two graphs with $n$ nodes and let $k \geq 1$. Let $f$ be a $k$-configuration mapping to tuple $\boldsymbol{u} \in V(G)^k$ and let $g$ be a $k$-configuration mapping to tuple $\boldsymbol{v} \in V(H)^k$. Then, for every $t \geq 0$,*

$$C_t^{k,F}(\boldsymbol{u}) = C_t^{k,F}(\boldsymbol{v}),$$

*if and only if $\boldsymbol{u}$ and $\boldsymbol{v}$ satisfy the same sentences in $\mathcal{C}_{k+1,t}$ whose free variables are in $\{x_1, x_2, \ldots, x_k\}$.*

Together with Theorem 1, we obtain the above results also for the embeddings of the ET for $k = 2$.

**Corollary 3.** *Let $G := (V(G), E(G))$ and $H := (V(H), E(H))$ be two graphs with $n$ nodes and let $k = 2$. Let $f$ be a $2$-configuration mapping to node pair $\boldsymbol{u} \in V(G)^2$ and let $g$ be a $2$-configuration mapping to node pair $\boldsymbol{v} \in V(H)^k$. Then, for every $t \geq 0$,*

$$\boldsymbol{X}^{(t)}(\boldsymbol{u}) = \boldsymbol{X}^{(t)}(\boldsymbol{v}),$$

*if and only if $\boldsymbol{u}$ and $\boldsymbol{v}$ satisfy the same sentences in $\mathcal{C}_{3,t}$ whose free variables are in $\{x_1, x_2\}$.*

**Systematic generalization** Returning to the example in Bergen et al. [6], the above result tells us that a model with 2-FWL expressive power and at least $t$ layers is equivalently able to evaluate sentences in $\mathcal{C}_{3,t}$, including

$$\mathrm{GRANDMOTHER}(x, z) = \exists y \big( \mathrm{MOTHER}(x, y) \wedge \mathrm{MOTHER}(y, z) \big),$$

i.e., the grandmother relation, and store this information encoded in some 2-tuple representation $\boldsymbol{X}^{(t)}(\boldsymbol{u})$, where $\boldsymbol{u} = (u, v)$ and $\boldsymbol{X}^{(t)}(\boldsymbol{u})$ encodes whether $u$ is a grandmother of $v$. As a result, we have theoretical justification for the intuitive argument made by Bergen et al. [6], namely that the ET can learn an *explicit* representation of a novel concept, in our example the GRANDMOTHER relation.

However, when closely examining the language $\mathcal{C}_{3,t}$, we find that the above result allows for an even wider theoretical justification of the systematic generalization ability of the ET. Concretely, we will show that once the ET obtains a representation for a novel concept such as the GRANDMOTHER relation, at some layer $t$, the ET can re-combine said concept to generalize to even more complex concepts. For example, consider the following relation, which we naively write as

$$\mathrm{GREATGRANDMOTHER}(x, a) = \exists z \exists y \big( \mathrm{MOTHER}(x, y) \wedge \mathrm{MOTHER}(y, z) \wedge \mathrm{MOTHER}(z, a) \big).$$

At first glance, it seems as though GREATGRANDMOTHER $\in \mathcal{C}_{4,2}$ but GREATGRANDMOTHER $\notin \mathcal{C}_{3,t}$ for any $t \geq 1$. However, notice that the variable $y$ serves merely as an intermediary to establish the GRANDMOTHER relation. Hence, we can, without loss of generality, write the above as

$$\text{GREATGRANDMOTHER}(x, a) = \exists y \underbrace{\left( \exists a \big( \text{MOTHER}(x, a) \wedge \text{MOTHER}(a, y) \big) \right)}_{a \text{ is re-quantified and temporarily bound}} \wedge \text{MOTHER}(y, a) \big),$$

i.e., we *re-quantify* $a$ to temporarily serve as the mother of $x$ and the daughter of $y$. Afterwards, $a$ is released and again refers to the great grandmother of $x$. As a result, GREATGRANDMOTHER $\in \mathcal{C}_{3,2}$ and hence the ET, as well as any other model with at least 2-FWL expressive power, is able to generalize to the GREATGRANDMOTHER relation within two layers, by iteratively re-combining existing concepts, in our example the GRANDMOTHER and the MOTHER relation. This becomes even more clear, by writing

$$\text{GREATGRANDMOTHER}(x, a) = \exists y \big( \text{GRANDMOTHER}(x, y) \wedge \text{MOTHER}(y, a) \big),$$

where GRANDMOTHER is a generalized concept obtained from the primitive concept MOTHER. To summarize, knowing the expressive power of a model such as the ET in terms of the Weisfeiler–Leman hierarchy allows us to draw direct connections to the logical reasoning abilities of the model. Further, this theoretical connection allows an interpretation of systematic generalization as the ability of a model with the expressive power of at least the $k$-FWL to iteratively re-combine concepts from first principles (such as the MOTHER relation) as a hierarchy of statements in $\mathcal{C}_{k+1,t}$, containing all FO-logic statements with counting quantifiers, at most $k + 1$ variables, and quantifier depth $t$.

## 6  Experimental evaluation

We now investigate how well the ET performs on various graph-learning tasks. We include tasks on graph-, node-, and edge-level. Specifically, we answer the following questions.

**Q1** How does the ET fare against other theoretically aligned architectures regarding predictive performance?

**Q2** How does the ET compare to state-of-the-art models?

**Q3** How effectively can the ET benefit from additional positional/structural encodings?

The source code for our experiments is available at `https://github.com/luis-mueller/towards-principled-gts`. To foster research in principled graph transformers such as the ET, we provide accessible implementations of ET, both in PyTorch and Jax.

**Datasets**  We evaluate the ET on graph-, node-, and edge-level tasks from various domains to demonstrate its versatility.

On the graph level, we evaluate the ET on the molecular datasets ZINC (12K), ZINC-FULL [14], ALCHEMY (12K), and PCQM4Mv2 [21]. Here, nodes represent atoms and edges bonds between atoms, and the task is always to predict one or more molecular properties of a given molecule. Due to their relatively small graphs, the above datasets are ideal for evaluating higher-order and other resource-hungry models.

On the node and edge level, we evaluate the ET on the CLRS benchmark for neural algorithmic reasoning [47]. Here, the input, output, and intermediate steps of 30 classical algorithms are translated into graph data, where nodes represent the algorithm input and edges are used to encode a partial ordering of the input. The algorithms of CLRS are typically grouped into eight algorithm classes: Sorting, Searching, Divide and Conquer, Greedy, Dynamic Programming, Graphs, Strings, and Geometry. The task is then to predict the output of an algorithm given its input. This prediction is made based on an encoder-processor-decoder framework introduced by Velickovic et al. [47], which is recursively applied to execute the algorithmic steps iteratively. We will use the ET as the processor in this framework, receiving as input the current algorithmic state in the form of node and edge features and outputting the updated node and edge features, according to the latest version of CLRS, available at `https://github.com/google-deepmind/clrs`. As such, the CLRS requires the ET to make both node- and edge-level predictions.

Finally, we conduct empirical expressivity tests on the BREC benchmark [49]. BREC contains 400 pairs of non-isomorphic graphs with up to 198 nodes, ranging from basic, 1-WL distinguishable

Table 1: Average test results and standard deviation for the molecular regression datasets. ALCHEMY (12K) and ZINC-FULL over 5 random seeds, ZINC (12K) over 10 random seeds.

| Model | ZINC (12K) | ALCHEMY (12K) | ZINC-FULL |
|---|---|---|---|
| | MAE $\downarrow$ | MAE $\downarrow$ | MAE $\downarrow$ |
| GIN(E) [51, 41] | 0.163 ±0.03 | 0.180 ±0.006 | 0.180 ±0.006 |
| CIN [8] | 0.079 ±0.006 | – | **0.022** ±0.002 |
| Graphormer-GD [54] | 0.081 ±0.009 | – | 0.025 ±0.004 |
| SignNet [30] | 0.084 ±0.006 | 0.113 ±0.002 | **0.024** ±0.003 |
| BasisNet [22] | 0.155 ±0.007 | 0.110 ±0.001 | – |
| PPGN++ [41] | 0.071 ±0.001 | 0.109 ±0.001 | **0.020** ±0.001 |
| SPE [22] | **0.069** ±0.004 | **0.108** ±0.001 | – |
| ET | **0.062** ±0.004 | **0.099** ±0.001 | 0.026 ±0.003 |
| ET+RRWP | **0.059** ±0.004 | **0.098** ±0.001 | **0.024** ±0.003 |

graphs to graphs even indistinguishable by 4-WL. In addition, BREC comes with its own training and evaluation pipeline. Let $f\colon \mathcal{G} \to \mathbb{R}^d$ be the model whose expressivity we want to test, where $f$ maps from a set of graphs $\mathcal{G}$ to $\mathbb{R}^d$ for some $d > 0$. Let $(G, H)$ be a pair of non-isomorphic graphs. During training, $f$ is trained to maximize the cosine distance between graph embeddings $f(G)$ and $f(H)$. During the evaluation, BREC decides whether $f$ can distinguish $G$ and $H$ by conducting a Hotelling's T-square test with the null hypothesis that $f$ cannot distinguish $G$ and $H$.

**Baselines** On the molecular regression datasets, we compare the ET to an 1-WL expressive GNN baseline such as GIN(E) [52].

On ZINC (12K), ZINC-FULL and ALCHEMY, we compare the ET to other theoretically-aligned models, most notably higher-order GNNs [8, 37, 39], Graphormer-GD, with strictly less expressive power than the 3-WL [54], and PPGN++, with strictly more expressive power than the 3-WL [41] to study **Q1**. On PCQM4Mv2, we compare the ET to state-of-the-art graph transformers to study **Q2**. To study the impact of positional/structural encodings in **Q3**, we evaluate the ET both with and without relative random walk probabilities (RRWP) positional encodings, recently proposed in Ma et al. [32]. RRWP encodings only apply to models with explicit representations over node pairs and are well-suited for the ET.

On the CLRS benchmark, we mostly compare to the Relational Transformer (RT) [12] as a strong graph transformer baseline. Comparing the ET to the RT allows us to study **Q2** in a different domain than molecular regression and on node- and edge-level tasks. Further, since the RT is similarly motivated as the ET in learning explicit representations of relations, we can study the potential benefits of the ET provable expressive power on the CLRS tasks. In addition, we compare the ET to DeepSet and GNN baselines in Diao and Loynd [12] and the single-task Triplet-GMPNN in Ibarz et al. [24].

On the BREC benchmark, we study questions **Q1** and **Q2** by comparing the ET to selected models presented in Wang and Zhang [49]. First, we compare to the $\delta$-2-LGNN [37], a higher-order GNN with strictly more expressive power than the 1-WL. Second, we compare to Graphormer [53], an empirically strong graph transformer. Third, we compare to PPGN [33] with the same expressive power as the ET. We additionally include the 3-WL results on the graphs in BREC to investigate how many 3-WL distinguishable graphs the ET can distinguish in BREC.

**Experimental setup** See Table 6 for an overview of the used hyperparameters.

For ZINC (12K), ZINC-FULL, and PCQM4Mv2, we follow the hyperparameters in Ma et al. [32]. For ALCHEMY, we follow standard protocol and split the data according to Morris et al. [39]. Here, we simply adopt the hyper-parameters of ZINC (12K) from Ma et al. [32] but set the batch size to 64.

We choose the same hyper-parameters as the RT for the CLRS benchmark. Also, following the RT, we train for 10K steps and report results over 20 random seeds. To stay as close as possible to the experimental setup of our baselines, we integrate our Jax implementation of the ET as a processor into the latest version of the CLRS code base. In addition, we explore the OOD validation technique presented in Jung and Ahn [25], where we use larger graphs for the validation set to encourage size

Table 2: Average test micro F1 of different algorithm classes and average test score of all algorithms in CLRS over ten random seeds; see Appendix B.3 for test scores per algorithm and Appendix B.4 for details on the standard deviation.

| Algorithm | Deep Sets [12] | GAT [12] | MPNN [12] | PGN [12] | RT [12] | Triplet-GMPNN [24] | ET (ours) |
|---|---|---|---|---|---|---|---|
| Sorting | **68.89** | 21.25 | 27.12 | 28.93 | 50.01 | 60.37 | 82.26 |
| Searching | 50.99 | 38.04 | 43.94 | 60.39 | 65.31 | 58.61 | 63.00 |
| DC | 12.29 | 15.19 | 16.14 | 51.30 | 66.52 | 76.36 | 64.44 |
| Greedy | 77.83 | 75.75 | **89.40** | 76.72 | 85.32 | 91.21 | 81.67 |
| DP | 68.29 | 63.88 | 68.81 | 71.13 | 83.20 | 81.99 | 83.49 |
| Graphs | 42.09 | 55.53 | 63.30 | 64.59 | 65.33 | 81.41 | 86.08 |
| Strings | 2.92 | 1.57 | 2.09 | 1.82 | 32.52 | 49.09 | 54.84 |
| Geometry | 65.47 | 68.94 | 83.03 | 67.78 | 84.55 | 94.09 | 88.22 |
| Avg. class | 48.60 | 41.82 | 49.23 | 52.83 | 66.60 | 74.14 | 75.51 |
| All algorithms | 50.29 | 48.08 | 55.15 | 56.57 | 66.18 | 75.98 | 80.13 |

generalization. This technique can be used within the CLRS code base through the experiment parameters.

Finally, for BREC, we keep the default hyper-parameters and follow closely the setup used by Wang and Zhang [49] for PPGN. We found learning on BREC to be quite sensitive to architectural choices, possibly due to the small dataset sizes. As a result, we use a linear layer for the FFN and additionally apply layer normalization onto $X_{il} W^Q$, $X_{lj} W^K$ in Equation (2) and $V_{ilj}$ in Equation (3).

For ZINC (12K), ZINC-FULL, PCQM4MV2, CLRS, and BREC, we follow the standard train/validation/test splits. For ALCHEMY, we split the data according to the splits in Morris et al. [39], the same as our baselines.

All experiments were performed on a mix of A10, L40, and A100 NVIDIA GPUs. For each run, we used at most 8 CPU cores and 64 GB of RAM, with the exception of PCQM4MV2 and ZINC-FULL, which were trained on 4 L40 GPUs with 16 CPU cores and 256 GB RAM.

Table 3: Number of distinguished pairs of non-isomorphic graphs on the BREC benchmark over 10 random seeds with standard deviation. Baseline results (over 1 random seed) are taken from Wang and Zhang [49]. For reference, we also report the number of graphs distinguishable by 3-WL.

| Model | Basic | Regular | Extension | CFI | *All* |
|---|---|---|---|---|---|
| $\delta$-2-LGNN | 60 | 50 | 100 | 6 | 216 |
| PPGN | 60 | 50 | 100 | 23 | 233 |
| Graphormer | 16 | 12 | 41 | 10 | 79 |
| ET | 60 ±0.0 | 50 ±0.0 | 100 ±0.0 | 48.1 ±1.9 | 258.1 ±1.9 |
| 3-WL | 60 | 50 | 100 | 60 | 270 |

**Results and discussion** In the following, we answer questions **Q1** to **Q3**. We highlight first, second, and third best results in each table.

We compare results on the molecular regression datasets in Table 1. On ZINC (12K) and ALCHEMY, the ET outperforms all baselines, even without using positional/structural encodings, positively answering **Q1**. Interestingly, on ZINC-FULL, the ET, while still among the best models, does not show superior performance. Further, the RRWP encodings we employ on the graph-level datasets improve the performance of the ET on all three datasets, positively answering **Q3**. Moreover, in Table 5, we compare the ET with a variety of graph learning models on ZINC (12K), demonstrating that the ET is highly competitive with state-of-the-art models. We observe similarly positive results in Table 4 where the ET outperforms strong graph transformer baselines such as GRIT [32], GraphGPS [42] and Graphormer [53] on PCQM4MV2. As a result, we can positively answer **Q2**.

Table 4: Average validation MAE on the PCQM4Mv2 benchmark over a single random seed.

| Model | Val. MAE ($\downarrow$) | # Params |
|---|---|---|
| EGT [23] | 0.0869 | 89.3M |
| GraphGPS$_{Small}$ [42] | 0.0938 | 6.2M |
| GraphGPS$_{Medium}$ [42] | 0.0858 | 19.4M |
| TokenGT$_{ORF}$ [28] | 0.0962 | 48.6M |
| TokenGT$_{Lap}$ [28] | 0.0910 | 48.5M |
| Graphormer [53] | 0.0864 | 48.3M |
| GRIT [32] | 0.0859 | 16.6M |
| GPTrans-L | 0.0809 | 86.0M |
| ET | 0.0840 | 16.8M |
| ET$_{+RRWP}$ | 0.0832 | 16.8M |

Table 5: ZINC (12K) leaderboard.

| Model | ZINC (12K) MAE $\downarrow$ |
|---|---|
| SignNet [30] | 0.084 ±0.006 |
| SUN [16] | 0.083 ±0.003 |
| Graphormer-GD [54] | 0.081 ±0.009 |
| CIN [8] | 0.079 ±0.006 |
| Graph-MLP-Mixer [20] | 0.073 ±0.001 |
| PPGN++ [41] | 0.071 ±0.001 |
| GraphGPS [42] | 0.070 ±0.004 |
| SPE [22] | 0.069 ±0.004 |
| Graph Diffuser [18] | 0.068 ±0.002 |
| Specformer [7] | 0.066 ±0.003 |
| GRIT [32] | 0.059 ±0.002 |
| ET | 0.062 ±0.004 |
| ET$_{+RRWP}$ | 0.059 ±0.004 |

In Table 2, we compare results on CLRS where the ET performs best when averaging all tasks or when averaging all algorithm classes, improving over RT and Triplet-GMPNN. Additionally, the ET performs best on 4 algorithm classes and is among the top 3 in 7/8 algorithm classes. Interestingly, only some models are best on a majority of algorithm classes. These results indicate a benefit of the ETs' expressive power on this benchmark, adding to the answer of **Q2**. Further, see Table 7 in Appendix B.2 for additional results using the OOD validation technique.

Finally, on the BREC benchmark, we observe that the ET cannot distinguish all graphs distinguishable by 3-WL. At the same time, the ET distinguishes more graphs than PPGN, the other 3-WL expressive model, providing an additional positive answer to **Q1**; see Table 3. Moreover, the ET distinguishes more graphs than $\delta$-2-LGNN and outperforms Graphormer by a large margin, again positively answering **Q2**. Overall, the positive results of the ET on BREC indicate that the ET is well able to leverage its expressive power empirically.

# 7    Limitations

While proving to be a strong and versatile graph model, the ET has an asymptotic runtime and memory complexity of $\mathcal{O}(n^3)$, which is more expensive than most state-of-the-art models with linear or quadratic runtime and memory complexity. We emphasize that due to the runtime and memory complexity of the $k$-WL, a trade-off between expressivity and efficiency is likely unavoidable. At the same time, the ET is highly parallelizable and runs efficiently on modern GPUs. We hope that innovations for parallelizable neural networks can compensate for the asymptotic runtime and memory complexity of the ET. In Figure 4 in the appendix, we find that we can use low-level GPU optimizations, available for parallelizable neural networks out-of-the-box, to dampen the cubic runtime and memory scaling of the ET; see Appendix C for runtime and memory experiments and an extended discussion.

# 8    Conclusion

We established a previously unknown connection between the Edge Transformer and 3-WL, and enabled the Edge Transformer for various graph learning tasks, including graph-, node-, and edge-level tasks. We also utilized a well-known connection between graph isomorphism testing and first-order logic to derive a theoretical interpretation of systematic generalization. We demonstrated empirically that the Edge Transformer is a promising architecture for graph learning, outperforming other theoretically aligned architectures and being among the best models on ZINC (12K), PCQM4Mv2 and CLRS. Furthermore, the ET is a graph transformer that does not rely on positional/structural encodings for strong empirical performance. Future work could further explore the potential of the Edge Transformer in neural algorithmic reasoning and molecular learning by improving its scalability to larger graphs, in particular through architecture-specific low-level GPU optimizations and model parallelism.

## Acknowledgments and Disclosure of Funding

CM and LM are partially funded by a DFG Emmy Noether grant (468502433) and RWTH Junior Principal Investigator Fellowship under Germany's Excellence Strategy. We thank Erik Müller for crafting the figures.

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

Table 6: Hyperparameters of the Edge Transformer across all datasets.

| Hyperparameter | ZINC(12K) | ALCHEMY | ZINC-FULL | CLRS | BREC | PCQM4Mv2 |
|---|---|---|---|---|---|---|
| Learning rate | 0.001 | 0.001 | 0.001 | 0.00025 | 0.0001 | 0.0002 |
| Grad. clip norm | 1.0 | 1.0 | 1.0 | 1.0 | – | 5.0 |
| Batch size | 32 | 64 | 256 | 4 | 16 | 256 |
| Optimizer | AdamW | Adam | AdamW | Adam | Adam | AdamW |
| Num. layers | 10 | 10 | 10 | 3 | 5 | 10 |
| Hidden dim. | 64 | 64 | 64 | 192 | 32 | 384 |
| Num. heads | 8 | 8 | 8 | 12 | 4 | 16 |
| Activation | GELU | GELU | GELU | ReLU | – | GELU |
| Pooling | SUM | SUM | SUM | – | – | SUM |
| RRWP dim. | 32 | 32 | 32 | – | – | 128 |
| Weight decay | 1e-5 | 1e-5 | 1e-5 | – | 0.0001 | 0.1 |
| Dropout | 0.0 | 0.0 | 0.0 | 0.0 | 0.0 | 0.1 |
| Attention dropout | 0.2 | 0.2 | 0.2 | 0.0 | 0.0 | 0.1 |
| # Steps | – | – | – | 10K | – | 2M |
| # Warm-up steps | – | – | – | 0 | – | 60K |
| # Epochs | 2K | 2K | 1K | – | 20 | – |
| # Warm-up epochs | 50 | 50 | 50 | – | 0 | – |
| # RRWP steps | 21 | 21 | 21 | – | – | 22 |

# A  Implementation details

Here, we present details about implementing the ET in practice.

## A.1  Node-level readout

In what follows, we propose a pooling method from node pairs to nodes, which allows us also to make predictions for node- and graph-level tasks. For each node $i \in V(G)$, we compute

$$\mathsf{ReadOut}(i) \coloneqq \sum_{j \in [n]} \rho_1\Big(\boldsymbol{X}_{ij}^{(L)}\Big) + \rho_2\Big(\boldsymbol{X}_{ji}^{(L)}\Big),$$

where $\rho_1, \rho_2$ are neural networks and $\boldsymbol{X}^{(L)}$ is the node pair tensor after $L$ ET layers. We apply $\rho_1$ to node pairs where node $i$ is at the first position and $\rho_2$ to node pairs where node $i$ is at the second position. We found that making such a distinction has positive impacts on empirical performance. Then, for graph-level predictions, we first compute node-level readout as above and then use common graph-level pooling functions such as `sum` and `mean` [51] or `set2seq` [48] on the resulting node representations. We use this readout method in our molecular regression experiments in Section 6.

# B  Experimental details

Table 6 gives an overview of selected hyper-parameters for all experiments.

See Appendix B.2 through Appendix B.4 for detailed results on the CLRS benchmark. Note that in the case of CLRS, we evaluate in the single-task setting where we train a new set of parameters for each concrete algorithm, initially proposed in CLRS, to be able to compare against graph transformers fairly. We leave the multi-task learning proposed in Ibarz et al. [24] for future work.

## B.1  Data source and license

ZINC (12K), ALCHEMY (12K) and ZINC-FULL are available at `https://pyg.org` under an MIT license. PCQM4Mv2 is available at `https://ogb.stanford.edu/docs/lsc/pcqm4mv2/` under a CC BY 4.0 license. The CLRS benchmark is available at `https://github.com/google-deepmind/clrs` under an Apache 2.0 license. The BREC benchmark is available at `https://github.com/GraphPKU/BREC` under an MIT license.

Table 7: Average test scores for the different algorithm classes and average test scores of all algorithms in CLRS **with the OOD validation technique** over 10 seeds; see Appendix B.3 for test scores per algorithm and Appendix B.4 for details on the standard deviation. Baseline results for Triplet-GMPNN and TEAM are taken from Jung and Ahn [25]. Results in %.

| Algorithm | Triplet-GMPNN | TEAM | ET (ours) |
|---|---|---|---|
| Sorting | 72.08 | 68.75 | **88.35** |
| Searching | 61.89 | 63.00 | **80.00** |
| DC | 65.70 | 69.79 | **74.70** |
| Greedy | 91.21 | **91.80** | 88.29 |
| DP | **90.08** | 83.61 | 84.69 |
| Graphs | 77.89 | 81.86 | **89.89** |
| Strings | 75.33 | **81.25** | 51.22 |
| Geometry | 88.02 | **94.03** | 89.68 |
| Avg. algorithm class | 77.48 | 79.23 | **80.91** |
| All algorithms | 78.00 | 79.82 | **85.01** |

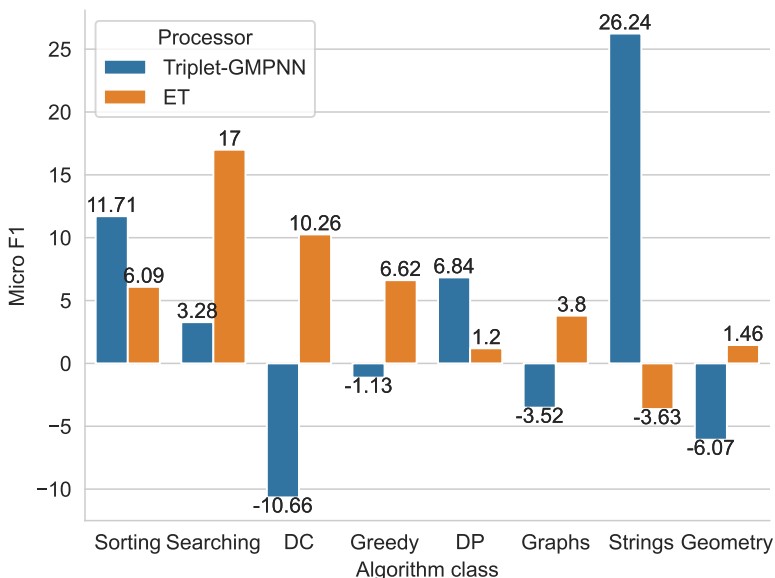

Figure 3: Difference in micro F1 with and without the OOD validation technique in Jung and Ahn [25], for Triplet-GMPNN [24] and ET, respectively.

## B.2 Experimental results OOD validation in CLRS

In Table 7, following [25], we present additional experimental results on CLRS when using graphs of size 32 in the validation set. We compare to both the Triplet-GMPNN [24], as well as the TEAM [25] baselines. In addition, in Figure 3, we present a comparison of the improvements resulting from the OOD validation technique, comparing Triplet-GMPNN and the ET. Finally, in Table 8, we compare different modifications to the CLRS training setup that are agnostic to the choice of processor.

## B.3 CLRS test scores

We present detailed results for the algorithms in CLRS. See Table 11 for divide and conquer algorithms, Table 12 for dynamic programming algorithms, Table 13 for geometry algorithms, Table 15 for greedy algorithms, Table 10 for search algorithms, Table 9 for sorting algorithms, and Table 16 for string algorithms.

Table 8: CLRS-30 Processor-agnostic modifications.

| Processor | Markov [9] | OOD Validation [25] | *Avg. algorithm class* | *All algorithms* |
|---|---|---|---|---|
| Triplet-GMPNN | ✓ | ✗ | **79.75** | **82.89** |
| Triplet-GMPNN | ✗ | ✓ | 77.65 | 78.00 |
| TEAM | ✗ | ✓ | 79.23 | 79.82 |
| ET | ✗ | ✓ | 80.91 | 85.02 |

Table 9: Detailed test scores for the ET on sorting algorithms.

| Algorithm | F1-score(%) | Std. dev.(%) | F1-score(%)(OOD) | Std. dev.(%) (OOD) |
|---|---|---|---|---|
| Bubble Sort | 93.60 | 3.87 | 87.44 | 13.48 |
| Heapsort | 64.36 | 22.41 | 80.96 | 12.97 |
| Insertion Sort | 85.71 | 20.68 | 91.74 | 6.83 |
| Quicksort | 85.37 | 8.70 | 93.25 | 9.10 |
| *Average* | 82.26 | 13.92 | 88.35 | 10.58 |

Table 10: Detailed test scores for the ET on search algorithms.

| Algorithm | F1-score(%) | Std. dev.(%) | F1-score(%)(OOD) | Std. dev.(%) (OOD) |
|---|---|---|---|---|
| Binary Search | 79.96 | 11.66 | 90.84 | 2.71 |
| Minimum | 96.88 | 1.74 | 97.94 | 0.87 |
| Quickselect | 12.43 | 11.72 | 52.64 | 22.04 |
| *Average* | 63.00 | 8.00 | 80.00 | 8.54 |

Table 11: Detailed test scores for the ET on divide and conquer algorithms.

| Algorithm | F1-score(%) | Std. dev.(%) | F1-score(%)(OOD) | Std. dev.(%) (OOD) |
|---|---|---|---|---|
| Find Max. Subarray Kadande | 64.44 | 2.24 | 74.70 | 2.59 |
| *Average* | 64.44 | 2.24 | 74.70 | 2.59 |

Table 12: Detailed test scores for the ET on dynamic programming algorithms.

| Algorithm | F1-score(%) | Std. dev.(%) | F1-score(%)(OOD) | Std. dev.(%) (OOD) |
|---|---|---|---|---|
| LCS Length | 88.67 | 2.05 | 88.97 | 2.06 |
| Matrix Chain Order | 90.11 | 3.28 | 90.84 | 2.94 |
| Optimal BST | 71.70 | 5.46 | 74.26 | 10.84 |
| *Average* | 83.49 | 3.60 | 84.68 | 5.28 |

Table 13: Detailed test scores for the ET on geometry algorithms.

| Algorithm | F1-score(%) | Std. dev.(%) | F1-score(%)(OOD) | Std. dev.(%) (OOD) |
|---|---|---|---|---|
| Graham Scan | 92.23 | 2.26 | 96.09 | 0.96 |
| Jarvis March | 89.09 | 8.92 | 95.18 | 1.46 |
| Segments Intersect | 83.35 | 7.01 | 77.78 | 1.16 |
| *Average* | 88.22 | 6.09 | 89.68 | 1.19 |

Table 14: Detailed test scores for the ET on graph algorithms.

| Algorithm | F1-score(%) | Std. dev.(%) | F1-score(%)(OOD) | Std. dev.(%) (OOD) |
|---|---|---|---|---|
| Articulation Points | 93.06 | 0.62 | 95.47 | 2.35 |
| Bellman-Ford | 89.96 | 3.77 | 95.55 | 1.65 |
| BFS | 99.77 | 0.30 | 99.95 | 0.08 |
| Bridges | 91.95 | 10.00 | 98.28 | 2.64 |
| DAG Shortest Paths | 97.63 | 0.85 | 98,43 | 0.65 |
| DFS | 65.60 | 17.98 | 57.76 | 14.54 |
| Dijkstra | 91.90 | 2.99 | 97.32 | 7.32 |
| Floyd-Warshall | 61.53 | 5.34 | 83.57 | 1.79 |
| MST-Kruskal | 84.06 | 2.14 | 87.21 | 1.45 |
| MST-Prim | 93.02 | 2.41 | 93.00 | 1.61 |
| SCC | 65.80 | 8.13 | 74.58 | 5.31 |
| Topological Sort | 98.74 | 2.24 | 97.53 | 2.31 |
| *Average* | 86.08 | 4.73 | 89.92 | 3.02 |

Table 15: Detailed test scores for the ET on greedy algorithms.

| Algorithm | F1-score(%) | Std. dev.(%) | F1-score(%)(OOD) | Std. dev.(%) (OOD) |
|---|---|---|---|---|
| Activity Selector | 80.12 | 12.34 | 91.72 | 2.35 |
| Task Scheduling | 83.21 | 0.30 | 84.85 | 2.83 |
| *Average* | 81.67 | 6.34 | 88.28 | 2.59 |

## B.4   CLRS test standard deviation

We compare the standard deviation of Deep Sets, GAT, MPNN, PGN, RT, and ET following the comparison in Diao and Loynd [12]. Table 17 compares the standard deviation over all algorithms in the CLRS benchmark. We observe that the ET has the lowest overall standard deviation. The table does not contain results for Triplet-GMPNN [24] since we do not have access to the test results for each algorithm on each seed that are necessary to compute the overall standard deviation. However, Table 18 compares the standard deviation per algorithm class between Triplet-GMPNN and the ET. We observe that Triplet-GMPNN and the ET have comparable standard deviations except for search and string algorithms, where Triplet-GMPNN has a much higher standard deviation than the ET.

## C   Runtime and memory

Here, we provide additional information on the runtime and memory requirements of the ET in practice. Specifically, in Figure 4, we provide runtime scaling of the ET with and without low-level GPU optimizations in PyTorch on an A100 GPU with `bfloat16` precision. We measure the time for the forward pass of a single layer of the ET on a single graph (batch size of 1) with $n \in \{100, 200, ..., 700\}$ nodes and average the runtime over 100 repeats. We sample random Erdős-Renyi graphs with edge probability $0.05$. We use an embedding dimension of 64 and two attention heads. We find that the automatic compilation into Triton [45], performed automatically by `torch.compile`, improves the runtime and memory scaling. Specifically, with `torch.compile` enabled, the ET layer can process graphs with up to 700 nodes and shows much more efficient runtime scaling with the number of nodes.

Table 16: Detailed test scores for the ET on string algorithms.

| Algorithm | F1-score(%) | Std. dev.(%) | F1-score(%)(OOD) | Std. dev.(%) (OOD) |
|---|---|---|---|---|
| KMP Matcher | 10.47 | 10.28 | 8.67 | 8.14 |
| Naive String Match | 99.21 | 1.10 | 93.76 | 6.28 |
| *Average* | 54.84 | 5.69 | 51.21 | 7.21 |

Table 17: Standard deviation of Deep Sets, GAT, MPNN, PGN, RT, and ET (over all algorithms and all seeds).

| Model | Std. Dev. (%) |
|---|---|
| Deep Sets | 29.3 |
| GAT | 32.3 |
| MPNN | 34.6 |
| PGN | 33.1 |
| RT | 29.6 |
| ET | **26.6** |

Table 18: Standard deviation per algorithm class of Triplet-GMPNN (over 10 random seeds) as reported in Ibarz et al. [24] and ET (over 10 random seeds). Results in %.

| Algorithm class | Triplet-GMPNN | ET |
|---|---|---|
| Sorting | 12.16 | 15.57 |
| Searching | 24.34 | 3.51 |
| Divide and Conquer | 1.34 | 2.46 |
| Greedy | 2.95 | 6.54 |
| Dynamic Programming | 4.98 | 3.60 |
| Graphs | 6.21 | 6.79 |
| Strings | 23.49 | 8.60 |
| Geometry | 2.30 | 3.77 |
| *Average* | 9.72 | **6.35** |

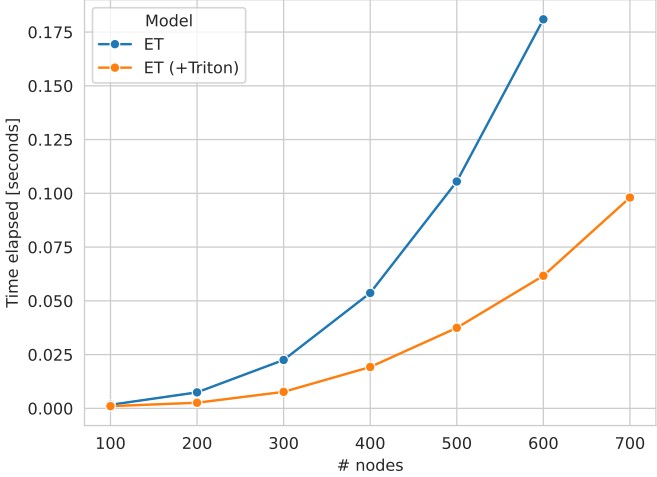

Figure 4: Runtime of the forward pass of a single ET layer in PyTorch in seconds for graphs with up to 700 nodes. We compare the runtime with and without `torch.compile` (automatic compilation into Triton [45]) enabled. Without compilation, the ET goes out of memory after 600 nodes.

Table 19: Runtime of a single run of the ET in CLRS on a single A100 GPU.

| Algorithm | Time in hh:mm:ss |
|---|---|
| Activity Selector | 00:09:38 |
| Articulation Points | 01:19:39 |
| Bellman Ford | 00:07:55 |
| BFS | 00:07:03 |
| Binary Search | 00:05:53 |
| Bridges | 01:20:44 |
| Bubble Sort | 01:05:34 |
| DAG Shortest Paths | 00:29:15 |
| DFS | 00:27:47 |
| Dijkstra | 00:09:37 |
| Find Maximum Subarray Kadane | 00:15:25 |
| Floyd Warshall | 00:12:56 |
| Graham Scan | 00:15:55 |
| Heapsort | 00:57:14 |
| Insertion Sort | 00:10:39 |
| Jarvis March | 01:34:40 |
| Kmp Matcher | 00:57:56 |
| LCS Length | 00:08:12 |
| Matrix Chain Order | 00:15:31 |
| Minimum | 00:21:25 |
| MST Kruskal | 01:15:54 |
| MST Prim | 00:09:34 |
| Naive String Matcher | 00:51:05 |
| Optimal BST | 00:12:57 |
| Quickselect | 02:25:03 |
| Quicksort | 00:59:24 |
| Segments Intersect | 00:03:38 |
| Strongly Connected Components | 00:56:58 |
| Task Scheduling | 00:08:50 |
| Topological Sort | 00:27:40 |

Table 20: Runtime of a single run on the molecular regression datasets, as well as BREC, on L40 GPUs in *days:hours:minutes:seconds*.

| | ZINC (12K) | ALCHEMY (12K) | ZINC-FULL | PCQM4MV2 | BREC |
|---|---|---|---|---|---|
| ET | 00:06:04:52 | 00:02:47:51 | 00:23:11:05 | 03:10:35:11 | 00:00:08:37 |
| ET+RRWP | 00:06:19:52 | 00:02:51:23 | 01:01:10:55 | 03:10:22:06 | - |
| Num. GPUs | 1 | 1 | 4 | 4 | 1 |

**Hardware optimizations**   Efficient compilation of neural networks is already available via programming languages such as Triton [45]. We use `torch.compile` in our molecular regression experiments. In addition, we want to highlight `FlashAttention` [11], available for the standard transformer, as an example of architecture-specific hardware optimizations that can reduce runtime and memory requirements.

**Runtime per dataset/benchmark**   Here, we present additional runtime results for all of our datasets. We present the runtime of a single run on a single L40 GPU of ZINC (12K), ALCHEMY (12K), and BREC. For ZINC-FULL and PCQM4Mv2, we present the runtime of a single run on 4 L40 GPUs; see Table 20.

On CLRS, the experiments in our work are run on a mix of A10 and A100 GPUs. To enable a fair comparison, we rerun each algorithm in CLRS in a single run on a single A100 GPU and report the corresponding runtime in Table 19. Finally, we note that these numbers only reflect the time to run the final experiments and significantly more time was used for preliminary experiments over the course of the research project.

# D   Extended preliminaries

Here, we define our notation. Let $\mathbb{N} := \{1, 2, 3, \dots\}$. For $n \geq 1$, let $[n] := \{1, \dots, n\} \subset \mathbb{N}$. We use $\{\!\{\dots\}\!\}$ to denote multisets, i.e., the generalization of sets allowing for multiple instances for each of its elements.

**Graphs**   A *(node-)labeled graph* $G$ is a triple $(V(G), E(G), \ell)$ with *finite* sets of *vertices* or *nodes* $V(G)$, *edges* $E(G) \subseteq \{\{u, v\} \subseteq V(G) \mid u \neq v\}$ and a (node-)label function $\ell \colon V(G) \to \mathbb{N}$. Then $\ell(v)$ is a *label* of $v$, for $v$ in $V(G)$. If not otherwise stated, we set $n := |V(G)|$, and the graph is of *order* $n$. We also call the graph $G$ an $n$-order graph. For ease of notation, we denote the edge $\{u, v\}$ in $E(G)$ by $(u, v)$ or $(v, u)$. We define an $n$-order *attributed graph* as a pair $\mathcal{G} = (G, \boldsymbol{F})$, where $G = (V(G), E(G))$ and $\boldsymbol{F}$ in $\mathbb{R}^{n \times p}$ for $p > 0$ is a *node feature matrix*. Here, we identify $V(G)$ with $[n]$, then $\boldsymbol{F}(v)$ in $\mathbb{R}^{1 \times p}$ is the *feature* or *attribute* of the node $v \in V(G)$. Given a labeled graph $(V(G), E(G), \ell)$, a node feature matrix $\boldsymbol{F}$ is *consistent* with $\ell$ if $\ell(v) = \ell(w)$ for $v, w \in V(G)$ if and only if $\boldsymbol{F}(v) = \boldsymbol{F}(w)$.

**Neighborhood and Isomorphism**   The *neighborhood* of a vertex $v$ in $V(G)$ is denoted by $N(v) := \{u \in V(G) \mid (v, u) \in E(G)\}$ and the *degree* of a vertex $v$ is $|N(v)|$. Two graphs $G$ and $H$ are *isomorphic* and we write $G \simeq H$ if there exists a bijection $\varphi \colon V(G) \to V(H)$ preserving the adjacency relation, i.e., $(u, v)$ is in $E(G)$ if and only if $(\varphi(u), \varphi(v))$ is in $E(H)$. Then $\varphi$ is an *isomorphism* between $G$ and $H$. In the case of labeled graphs, we additionally require that $l(v) = l(\varphi(v))$ for $v$ in $V(G)$, and similarly for attributed graphs. Moreover, we call the equivalence classes induced by $\simeq$ *isomorphism types* and denote the isomorphism type of $G$ by $\tau_G$. We further define the atomic type $\mathrm{atp} \colon V(G)^k \to \mathbb{N}$, for $k > 0$, such that $\mathrm{atp}(\boldsymbol{v}) = \mathrm{atp}(\boldsymbol{w})$ for $\boldsymbol{v}$ and $\boldsymbol{w}$ in $V(G)^k$ if and only if the mapping $\varphi \colon V(G)^k \to V(G)^k$ where $v_i \mapsto w_i$ induces a partial isomorphism, i.e., we have $v_i = v_j \iff w_i = w_j$ and $(v_i, v_j) \in E(G) \iff (\varphi(v_i), \varphi(v_j)) \in E(G)$.

**Matrices**   Let $\boldsymbol{M} \in \mathbb{R}^{n \times p}$ and $\boldsymbol{N} \in \mathbb{R}^{n \times q}$ be two matrices then $\begin{bmatrix} \boldsymbol{M} & \boldsymbol{N} \end{bmatrix} \in \mathbb{R}^{n \times (p+q)}$ denotes column-wise matrix concatenation. We also write $\mathbb{R}^d$ for $\mathbb{R}^{1 \times d}$. Further, let $\boldsymbol{M} \in \mathbb{R}^{p \times n}$ and $\boldsymbol{N} \in \mathbb{R}^{q \times n}$ be two matrices then

$$\begin{bmatrix} \boldsymbol{M} \\ \boldsymbol{N} \end{bmatrix} \in \mathbb{R}^{(p+q) \times n}$$

denotes row-wise matrix concatenation.

For a matrix $\boldsymbol{X} \in \mathbb{R}^{n \times d}$, we denote with $\boldsymbol{X}_i$ the $i$th row vector. In the case where the rows of $\boldsymbol{X}$ correspond to nodes in a graph $G$, we use $\boldsymbol{X}_v$ to denote the row vector corresponding to the node $v \in V(G)$.

**The Weisfeiler–Leman algorithm**   We describe the Weisfeiler–Leman algorithm, starting with the 1-WL. The 1-WL or color refinement is a well-studied heuristic for the graph isomorphism problem,

originally proposed by Weisfeiler and Leman [50].[1] Intuitively, the algorithm determines if two graphs are non-isomorphic by iteratively coloring or labeling vertices. Formally, let $G = (V, E, \ell)$ be a labeled graph, in each iteration, $t > 0$, the 1-WL computes a vertex coloring $C_t^1 \colon V(G) \to \mathbb{N}$, depending on the coloring of the neighbors. That is, in iteration $t > 0$, we set

$$C_t^1(v) \coloneqq \mathsf{recolor}\Big(\big(C_{t-1}^1(v), \{\!\!\{ C_{t-1}^1(u) \mid u \in N(v) \}\!\!\}\big)\Big),$$

for all vertices $v$ in $V(G)$, where recolor injectively maps the above pair to a unique natural number, which has not been used in previous iterations. In iteration 0, the coloring $C_0^1 \coloneqq \ell$. To test if two graphs $G$ and $H$ are non-isomorphic, we run the above algorithm in "parallel" on both graphs. If the two graphs have a different number of vertices colored $c$ in $\mathbb{N}$ at some iteration, the 1-WL *distinguishes* the graphs as non-isomorphic. It is easy to see that 1-WL cannot distinguish all non-isomorphic graphs [10].

**The $k$-dimensional Weisfeiler–Leman algorithm**  Due to the shortcomings of the 1-WL or color refinement in distinguishing non-isomorphic graphs, several researchers, e.g., Babai [3], Cai et al. [10], devised a more powerful generalization of the former, today known as the $k$-dimensional Weisfeiler-Leman algorithm ($k$-WL), operating on $k$-tuples of nodes rather than single nodes.

Intuitively, to surpass the limitations of the 1-WL, the $k$-WL colors node-ordered $k$-tuples instead of a single node. More precisely, given a graph $G$, the $k$-WL colors the tuples from $V(G)^k$ for $k \geq 2$ instead of the nodes. By defining a neighborhood between these tuples, we can define a coloring similar to the 1-WL. Formally, let $G$ be a graph, and let $k \geq 2$. In each iteration, $t \geq 0$, the algorithm, similarly to the 1-WL, computes a *coloring* $C_t^k \colon V(G)^k \to \mathbb{N}$. In the first iteration, $t = 0$, the tuples $\boldsymbol{v}$ and $\boldsymbol{w}$ in $V(G)^k$ get the same color if they have the same atomic type, i.e., $\mathrm{atp}_k(\boldsymbol{v}) = \mathrm{atp}_k(\boldsymbol{u})$. Then, for each iteration, $t > 0$, $C_t^k$ is defined by

$$C_t^k(\boldsymbol{v}) \coloneqq \mathsf{recolor}\big(C_{t-1}^k(\boldsymbol{v}), M_t(\boldsymbol{v})\big), \tag{5}$$

with $M_t(\boldsymbol{v})$ the multiset

$$M_t(\boldsymbol{v}) \coloneqq \big(\{\!\!\{ C_{t-1}^k(\phi_1(\boldsymbol{v}, w)) \mid w \in V(G) \}\!\!\}, \ldots, \{\!\!\{ C_{t-1}^k(\phi_k(\boldsymbol{v}, w)) \mid w \in V(G) \}\!\!\}\big), \tag{6}$$

and where

$$\phi_j(\boldsymbol{v}, w) \coloneqq (v_1, \ldots, v_{j-1}, w, v_{j+1}, \ldots, v_k).$$

That is, $\phi_j(\boldsymbol{v}, w)$ replaces the $j$-th component of the tuple $\boldsymbol{v}$ with the node $w$. Hence, two tuples are *adjacent* or *$j$-neighbors* if they are different in the $j$th component (or equal, in the case of self-loops). Hence, two tuples $\boldsymbol{v}$ and $\boldsymbol{w}$ with the same color in iteration $(t-1)$ get different colors in iteration $t$ if there exists a $j$ in $[k]$ such that the number of $j$-neighbors of $\boldsymbol{v}$ and $\boldsymbol{w}$, respectively, colored with a certain color is different.

We run the $k$-WL algorithm until convergence, i.e., until for $t$ in $\mathbb{N}$

$$C_t^k(\boldsymbol{v}) = C_t^k(\boldsymbol{w}) \iff C_{t+1}^k(\boldsymbol{v}) = C_{t+1}^k(\boldsymbol{w}),$$

for all $\boldsymbol{v}$ and $\boldsymbol{w}$ in $V(G)^k$ holds.

Similarly to the 1-WL, to test whether two graphs $G$ and $H$ are non-isomorphic, we run the $k$-WL in "parallel" on both graphs. Then, if the two graphs have a different number of nodes colored $c$, for $c$ in $\mathbb{N}$, the $k$-WL *distinguishes* the graphs as non-isomorphic. By increasing $k$, the algorithm gets more powerful in distinguishing non-isomorphic graphs, i.e., for each $k \geq 2$, there are non-isomorphic graphs distinguished by $(k+1)$-WL but not by $k$-WL [10].

**The folklore $k$-dimensional Weisfeiler–Leman algorithm**  A common and well-studied variant of the $k$-WL is the $k$-FWL, which differs from the $k$-WL only in the aggregation function. Instead of Equation (6), the "folklore" version of the $k$-WL updates $k$-tuples according to

$$M_t^{\mathrm{F}}(\boldsymbol{v}) \coloneqq \{\!\!\{ (C_{t-1}^{k,\mathrm{F}}(\phi_1(\boldsymbol{v}, w)), \ldots, C_{t-1}^{k,\mathrm{F}}(\phi_k(\boldsymbol{v}, w))) \mid w \in V(G) \}\!\!\},$$

resulting in the coloring $C_t^{k,\mathrm{F}} \colon V(G)^k \to \mathbb{N}$, and is strictly more powerful than the $k$-WL. Specifically, for $k \geq 2$, the $k$-WL is exactly as powerful as the $(k-1)$-FWL [19].

---

[1]Strictly speaking, the 1-WL and color refinement are two different algorithms. The 1-WL considers neighbors and non-neighbors to update the coloring, resulting in a slightly higher expressive power when distinguishing vertices in a given graph; see [19] for details. For brevity, we consider both algorithms to be equivalent.

**Computing $k$-WL's initial colors** Let $G = (V(G), E(G), \ell)$ be a labeled graph, $k \geq 2$, and let $\boldsymbol{v} := (v_1, \ldots, v_k) \in V(G)^k$ be a $k$-tuple. Then, we can present the atomic type $\mathrm{atp}(\boldsymbol{v})$ by a $k \times k$ matrix $K$ over $\{1, 2, 3\}$. That is, the entry $K_{ij}$ is 1 if $(v_i, v_j) \in E(G)$, 2 if $v_i = v_j$, and 3 otherwise. Further, we ensure consistency with $\ell$, meaning that for two $k$-tuples $\boldsymbol{v} := (v_1, \ldots, v_k) \in V(G)^k$ and $\boldsymbol{w} := (w_1, \ldots, w_k) \in V(G)^k$, then

$$C_0^k(\mathbf{v}) = C_0^k(\mathbf{w}),$$

if and only if, $\mathrm{atp}(\mathbf{v}) = \mathrm{atp}(\mathbf{w})$ and $\ell(v_i) = \ell(w_i)$, for all $i \in [k]$. Note that we compute the initial colors for both $k$-WL and the $k$-FWL in this way.

### D.1 Relationship between first-order logic and Weisfeiler–Leman

We begin with a short review of Cai et al. [10]. We consider our usual node-labeled graph $G = (V(G), E(G), \ell)$ with $n$ nodes. However, we replace $\ell$ with a countable set of color relations $C_1, \ldots, C_n$, where for a node $v \in V(G)$,

$$C_i(v) \Longleftrightarrow \ell(v) = i.$$

Note that Cai et al. [10] consider the more general case where nodes can be assigned to multiple colors simultaneously. However, for our work, we assume that a node is assigned to precisely one color, and hence, the set of color relations is at most of size $n$. We can construct first-order logic statements about $G$. For example, the following sentence describes the existence of a triangle formed by two nodes with color 1:

$$\exists x_1 \exists x_2 \exists x_3 \big( E(x_1, x_2) \wedge E(x_1, x_3) \wedge E(x_2, x_3) \wedge C_1(x_1) \wedge C_1(x_2) \big).$$

Here, $x_1$, $x_2$, and $x_3$ are *variables* which can be repeated and re-quantified at will. Statements made about $G$ and a subset of nodes in $V(G)$ are of particular importance to us. To this end, we define a *$k$-configuration*, a function $f : \{x_1, \ldots, x_k\} \to V(G)$ that assigns a node in $V(G)$ to each one of the variables $x_1, \ldots, x_k$. Let $\varphi$ be a first-order formula with free variables among $x_1, \ldots, x_k$. Then, we write

$$G, f \models \varphi$$

if $\varphi$ is true when the variable $x_i$ is interpreted as the node $f(x_i)$, for $i = 1, \ldots, k$.

Cai et al. [10] define the language $\mathcal{C}_{k,m}$ of all first-order formulas with counting quantifiers, at most $k$ variables, and quantifier depth bounded by $m$, and the language $\mathcal{C}_k = \bigcup_{m \geq 0} \mathcal{C}_{k,m}$. For example, the sentence $\forall x \exists! 3y \big( E(x, y) \big)$ in $\mathcal{C}_2$ describes 3-regular graphs; i.e., graphs where each vertex has exactly 3 neighbors.

We define the equivalence relation $\equiv_{k,m}$ over pairs $(G, f)$ made of graphs $G$ and $k$-configurations $f$ as $(G, f) \equiv_{k,m} (H, g)$ if and only if

$$G, f \models \varphi \iff H, g \models \varphi$$

for all formulas $\varphi$ in $\mathcal{C}_{k,m}$ whose free variables are among $x_1, \ldots, x_k$.

We can now formulate a main result of Cai et al. [10]. Let $G$ and $H$ be two graphs, let $k \geq 1$ and $m \geq 0$ be non-negative integers, and let $f$ and $g$ be $k$-configurations for $G$ and $H$ respectively. If $\boldsymbol{u} = (f(x_1), \ldots, f(x_k)) \in V(G)^k$ and $\boldsymbol{v} = (g(x_1), \ldots, g(x_k)) \in V(H)^k$, then

$$C_m^{k,F}(\boldsymbol{u}) = C_m^{k,F}(\boldsymbol{v}) \iff (G, f) \equiv_{k,m} (H, g).$$

## E Proofs

Here, we first generalize the GNN from Grohe [19] to the 2-FWL. Higher-order GNNs with the same expressivity have been proposed in prior works by Azizian and Lelarge [1]. However, our GNNs have a special form that can be computed by the Edge Transformer.

Formally, let $S \subseteq \mathbb{N}$ be a finite subset. First, we show that multisets over $S$ can be injectively mapped to a value in the closed interval $(0, 1)$, a variant of Lemma VIII.5 in Grohe [19]. Here, we outline a streamlined version of its proof, highlighting the key intuition behind representing multisets as $m$-ary numbers. Let $M \subseteq S$ be a multiset with multiplicities $a_1, \ldots, a_k$ and distinct $k$ values. We define the *order* of the multiset as $\sum_{i=1}^k a_i$. We can write such a multiset as a sequence $x^{(1)}, \ldots, x^{(l)}$ where

$l$ is the order of the multiset. Note that the order of the sequence is arbitrary and that for $i \neq j$ it is possible to have $x^{(i)} = x^{(j)}$. We call such a sequence an $M$-sequence of length $l$. We now prove a slight variation of a result of Grohe [19].

**Lemma 4.** *For a finite $m \in \mathbb{N}$, let $M \subseteq S$ be a multiset of order $m - 1$ and let $x_i \in S$ denote the ith number in a fixed but arbitrary ordering of $S$. Given a mapping $g \colon S \to (0, 1)$ where*

$$g(x_i) \coloneqq m^{-i},$$

*and an $M$-sequence of length $l$ given by $x^{(1)}, \ldots, x^{(l)}$ with positions $i^{(1)}, \ldots, i^{(l)}$ in $S$, the sum*

$$\sum_{j \in [l]} g(x^{(j)}) = \sum_{j \in [l]} m^{-i^{(j)}}$$

*is unique for every unique $M$.*

*Proof.* By assumption, let $M \subseteq S$ denote a multiset of order $m - 1$. Further, let $x^{(1)}, \ldots, x^{(l)} \in M$ be an $M$-sequence with $i^{(1)}, \ldots, i^{(l)}$ in $S$. Given our fixed ordering of the numbers in $S$ we can equivalently write $M = ((a_1, x_1), \ldots, (a_n, x_n))$, where $a_i$ denotes the multiplicity of $i$th number in $M$ with position $i$ from our ordering over $S$. Note that for a number $m^{-i}$ there exists a corresponding $m$-ary number written as

$$0.0\ldots\underbrace{1}_{i}\ldots$$

Then the sum,

$$\sum_{j \in [l]} g(x^{(j)}) = \sum_{j \in [l]} m^{-i^{(j)}}$$

$$= \sum_{i \in S} a_i m^{-i} \in (0, 1)$$

and in $m$-ary representation

$$0.a_1 \ldots a_n.$$

Note that $a_i = 0$ if and only if there exists no $j$ such that $i^{(j)} = i$. Since the order of $M$ is $m - 1$, it holds that $a_i < m$. Hence, it follows that the above sum is unique for each unique multiset $M$, implying the result. $\square$

Recall that $S \subseteq \mathbb{N}$ and that we fixed an arbitrary ordering over $S$. Intuitively, we use the finiteness of $S$ to map each number therein to a fixed digit of the numbers in $(0, 1)$. The finite $m$ ensures that at each digit, we have sufficient "bandwidth" to encode each $a_i$. Now that we have seen how to encode multisets over $S$ as numbers in $(0, 1)$, we review some fundamental operations about the $m$-ary numbers defined above. We will refer to decimal numbers $m^{-i}$ as *corresponding* to an $m$-ary number

$$0.0\ldots\underbrace{1}_{i}\ldots,$$

where the $i$th digit after the decimal point is 1 and all other digits are 0, and vice versa.

To begin with, addition between decimal numbers implements *counting* in $m$-ary notation, i.e.,

$$m^{-i} + m^{-j} \text{ corresponds to } 0.0\ldots\underbrace{1}_{i}\ldots\underbrace{1}_{j}\ldots,$$

for digit positions $i \neq j$ and

$$m^{-i} + m^{-j} \text{ corresponds to } 0.0\ldots\underbrace{2}_{i=j}\ldots,$$

otherwise. We used counting in the previous result's proof to represent a multiset's multiplicities. Next, multiplication between decimal numbers implements *shifting* in $m$-ary notation, i.e.,

$$m^{-i} \cdot m^{-j} \text{ corresponds to } 0.0\ldots\underbrace{1}_{i+j}\ldots.$$

Shifting further applies to general decimal numbers in $(0, 1)$. Let $x \in (0, 1)$ correspond to an $m$-ary number with $l$ digits,

$$0.a_1 \ldots a_l.$$

Then,

$$m^{-i} \cdot x \text{ corresponds to } 0.0 \ldots 0 \underbrace{a_1 \ldots a_l}_{i+1,\ldots,i+l}.$$

Before we continue, we show a small lemma stating that two non-overlapping sets of $m$-ary numbers preserve their uniqueness under addition.

**Lemma 5.** *Let $A$ and $B$ be two sets of $m$-ary numbers for some $m > 1$. If*

$$\min_{x \in A} x > \max_{y \in B} y,$$

*then for any $x_1, x_2 \in A, y_1, y_2 \in B$,*

$$x_1 + y_1 = x_2 + y_2 \iff x_1 = x_2 \text{ and } y_1 = y_2.$$

*Proof.* The statement follows from the fact that if

$$\min_{x \in A} x > \max_{y \in B} y,$$

then numbers in $A$ and numbers in $B$ do not overlap in terms of their digit range. Specifically, there exists some $l > 0$ such that we can write

$$x := 0.x_1 \ldots x_l$$
$$y := 0.\underbrace{0 \ldots 0}_{l} y_1 \ldots y_k,$$

for some $k > l$ and all $x \in A$, $y \in B$. As a result,

$$x + y = 0.x_1 \ldots x_l y_1 \ldots y_k.$$

Hence, $x + y$ is unique for every unique pair $(x, y)$. This completes the proof. $\square$

We begin by showing the following proposition, showing that the tokenization in Equation (4) is sufficient to encode the initial node colors under 2-FWL.

**Proposition 6.** *Let $G = (V(G), E(G), \ell)$ be a node-labeled graph with $n$ nodes. Then, there exists a parameterization of Equation (4) with $d = 1$ such that for each 2-tuples $\boldsymbol{u}, \boldsymbol{v} \in V(G)^2$,*

$$C_0^{2,F}(\boldsymbol{u}) = C_0^{2,F}(\boldsymbol{v}) \iff \boldsymbol{X}(\boldsymbol{u}) = \boldsymbol{X}(\boldsymbol{v}).$$

*Proof.* The statement directly follows from the fact that the initial color of a tuple $\boldsymbol{u} := (i, j)$ depends on the atomic type and the node labeling. In Equation (4), we encode the atomic type with $\boldsymbol{E}_{ij}$ and the node labels with

$$[\boldsymbol{E}_{ij} \quad \boldsymbol{F}_i \quad \boldsymbol{F}_j]$$

The concatenation of both node labels and atomic type is clearly injective. Finally, since there are at most $n^2$ distinct initial colors of the 2-FWL, said colors can be well represented within $\mathbb{R}$, hence there exists an injective $\phi$ in Equation (4) with $d = 1$. This completes the proof. $\square$

We now show Theorem 1. Specifically, we show the following two propositions from which Theorem 1 follows.

**Proposition 7.** *Let $G = (V(G), E(G), \ell)$ be a node-labeled graph with $n$ nodes and $\boldsymbol{F} \in \mathbb{R}^{n \times p}$ be a node feature matrix consistent with $\ell$. Then for all $t \geq 0$, there exists a parametrization of the ET such that*

$$C_t^{2,F}(\boldsymbol{v}) = C_t^{2,F}(\boldsymbol{w}) \impliedby \boldsymbol{X}^{(t)}(\boldsymbol{v}) = \boldsymbol{X}^{(t)}(\boldsymbol{w}),$$

*for all pairs of 2-tuples $\boldsymbol{v}$ and $\boldsymbol{w} \in V(G)^2$.*

*Proof.* We begin by stating that our domain is compact since the ET merely operates on at most $n$ possible node features in $\boldsymbol{F}$ and binary edge features in $\boldsymbol{E}$, and at each iteration there exist at most $n^2$ distinct 2-FWL colors. We prove our statement by induction over iteration $t$. For the base case, we can simply invoke Proposition 6 since our input tokens are constructed according to Equation (4). Nonetheless, we show a possible initialization of the tokenization that is consistent with Equation (4) that we will use in the induction step.

From Proposition 6, we know that the color representation of a tuple can be represented in $\mathbb{R}$. We denote the color representation of a tuple $\boldsymbol{u} = (i, j)$ at iteration $t$ as $\boldsymbol{T}^{(t)}(\boldsymbol{u})$ and $\boldsymbol{T}_{ij}^{(t)}$ interchangeably. We choose a $\phi$ in Equation (4) such that for each $\boldsymbol{u} = (i, j)$

$$\boldsymbol{X}_{ij}^{(0)} = \left[ \boldsymbol{T}_{ij}^{(0)} \quad \left( \boldsymbol{T}_{ij}^{(0)} \right)^{n^2} \right] \in \mathbb{R}^2,$$

where we store the tuple features, one with exponent 1 and once with exponent $n^2$ and where $\boldsymbol{T}_{ij}^{(0)} \in \mathbb{R}$ and $\left( \boldsymbol{T}_{ij}^{(0)} \right)^{n^2} \in \mathbb{R}$. We choose color representations $\boldsymbol{T}_{ij}^{(0)}$ as follows. First, we define an injective function $f_t : V(G)^2 \to [n^2]$ that maps each 2-tuple $\boldsymbol{u}$ to a number in $[n^2]$ unique for its 2-FWL color $C_t^{2,\mathrm{F}}(\boldsymbol{u})$ at iteration $t$. Note that $f_t$ can be injective because there can at most be $[n^2]$ unique numbers under the 2-FWL. We will use $f_t$ to map each tuple color under the 2-FWL to a unique $n$-ary number. We then choose $\phi$ in Equation (4) such that for each $(i, j) \in V(G)^2$,

$$\left\| \boldsymbol{T}_{ij}^{(0)} - n^{-f_0(i,j)} \right\|_F < \epsilon_0,$$

for all $\epsilon_0 > 0$, by the universal function approximation theorem, which we can invoke since our domain is compact. We will use $\left( \boldsymbol{T}_{ij}^{(0)} \right)^{n^2}$ in the induction step; see below.

For the induction, we assume that

$$C_{t-1}^{2,\mathrm{F}}(\boldsymbol{v}) = C_{t-1}^{2,\mathrm{F}}(\boldsymbol{w}) \Longleftarrow \boldsymbol{T}^{(t-1)}(\boldsymbol{v}) = \boldsymbol{T}^{(t-1)}(\boldsymbol{w})$$

and that

$$\left\| \boldsymbol{T}_{ij}^{(t-1)} - n^{-f_{t-1}(i,j)} \right\|_F < \epsilon_{t-1},$$

for all $\epsilon_{t-1} > 0$ and $(i, j) \in V(G)^2$. We then want to show that there exists a parameterization of the $t$-th layer such that

$$C_t^{2,\mathrm{F}}(\boldsymbol{v}) = C_t^{2,\mathrm{F}}(\boldsymbol{w}) \Longleftarrow \boldsymbol{T}^{(t)}(\boldsymbol{v}) = \boldsymbol{T}^{(t)}(\boldsymbol{w}) \tag{7}$$

and that

$$\left\| \boldsymbol{T}_{ij}^{(t)} - n^{-f_t(i,j)} \right\|_F < \epsilon_t,$$

for all $\epsilon_t > 0$ and $(i, j) \in V(G)^2$. Clearly, if this holds for all $t$, then the proof statement follows. Thereto, we show that the ET updates the tuple representation of tuple $(j, m)$ as

$$\boldsymbol{T}_{jm}^{(t)} = \mathsf{FFN}\left( \boldsymbol{T}_{jm}^{(t-1)} + \frac{\beta}{n} \sum_{l=1}^{n} \boldsymbol{T}_{jl}^{(t-1)} \cdot \left( \boldsymbol{T}_{lm}^{(t-1)} \right)^{n^2} \right), \tag{8}$$

for an arbitrary but fixed $\beta$. We first show that then, Equation (7) holds. Afterwards we show that the ET can indeed compute Equation (8). To show the former, note that for two 2-tuples $(j, l)$ and $(l, m)$,

$$n^{-n^2} \cdot n^{-f_{t-1}(j,l)} \cdot \left( n^{-f_{t-1}(l,m)} \right)^{n^2} = n^{-(n^2 + f_{t-1}(j,l) + n^2 \cdot f_{t-1}(l,m))},$$

is unique for the pair of colors

$$\left( C_t^{2,\mathrm{F}}((j, l)), C_t^{2,\mathrm{F}}((l, m)) \right)$$

where $n^{-n^2}$ is a constant normalization term we will later introduce with $\frac{\beta}{n}$. Note further, that we have

$$\left\| \boldsymbol{T}_{jl}^{(t-1)} \cdot \left( \boldsymbol{T}_{lm}^{(t-1)} \right)^{n^2} - n^{-(n^2 + f_{t-1}(j,l) + n^2 \cdot f_{t-1}(l,m))} \right\|_F < \delta_{t-1},$$

for all $\delta_{t-1} > 0$. Further, $n^{-(f_{t-1}(j,l) + n^2 \cdot f_{t-1}(l,m))}$ is still an $m$-ary number with $m = n$. As a result, we can set $\beta = n^{-n^2+1}$ and invoke Lemma 4 to obtain that

$$\frac{\beta}{n} \cdot \sum_{l=1}^{n} n^{-(f_{t-1}(j,l) + n^2 \cdot f_{t-1}(l,m))} = \sum_{l=1}^{n} n^{-(n^2 + f_{t-1}(j,l) + n^2 \cdot f_{t-1}(l,m))},$$

is unique for the multiset of colors

$$\{\!\!\{(C^{2,\text{F}}_{t-1}((l,m)), C^{2,\text{F}}_{t-1}((j,l))) \mid l \in V(G)\}\!\!\},$$

and we have that

$$\Big|\Big|\frac{\beta}{n}\sum_{l=1}^{n} \boldsymbol{T}^{(t-1)}_{jl} \cdot \Big(\boldsymbol{T}^{(t-1)}_{lm}\Big)^{n^2} - \sum_{l=1}^{n} n^{-(n^2+f_{t-1}(j,l)+n^2 \cdot f_{t-1}(l,m))}\Big|\Big|_F < \gamma_{t-1},$$

for all $\gamma_{t-1} > 0$. Finally, we define

$$A := \Big\{ n^{-f_{t-1}(j,m)} \mid (j,m) \in V(G)^2 \Big\}$$

$$B := \Big\{ \frac{\beta}{n} \cdot \sum_{l=1}^{n} n^{-(f_{t-1}(j,l)+n^2 \cdot f_{t-1}(l,m))} \mid (j,m) \in V(G)^2 \Big\}.$$

Further, because we multiply with $\frac{\beta}{n}$, we have that

$$\min_{x \in A} x > \max_{y \in B} y$$

and as a result, by Lemma 5,

$$n^{-f_{t-1}(j,m)} + \frac{\beta}{n} \cdot \sum_{l=1}^{n} n^{-(f_{t-1}(j,l)+n^2 \cdot f_{t-1}(l,m))}$$

is unique for the pair

$$\Big(C^{2,\text{F}}_{t-1}((j,m)), \{\!\!\{(C^{2,\text{F}}_{t-1}((l,m)), C^{2,\text{F}}_{t-1}((j,l))) \mid l \in V(G)\}\!\!\}\Big)$$

and consequently for color $C^{2,\text{F}}_t((j,m))$ at iteration $t$. Further, we have that

$$\Big|\Big|\boldsymbol{T}^{(t-1)}_{jm} + \frac{\beta}{n}\sum_{l=1}^{n} \boldsymbol{T}^{(t-1)}_{jl} \cdot \Big(\boldsymbol{T}^{(t-1)}_{lm}\Big)^{n^2} - n^{-f_{t-1}(j,m)} + \frac{\beta}{n} \cdot \sum_{l=1}^{n} n^{-(f_{t-1}(j,l)+n^2 \cdot f_{t-1}(l,m))}\Big|\Big|_F < \tau_{t-1},$$

for all $\tau_{t-1} > 0$. Finally, since our domain is compact, we can invoke universal function approximation with FFN in Equation (8) to obtain

$$\Big|\Big|\boldsymbol{T}^{(t)}_{jm} - n^{-f_t(j,m)}\Big|\Big|_F < \epsilon_t,$$

for all $\epsilon_t > 0$. Further, because $n^{-f_t(j,m)}$ is unique for each unique color $C^{2,\text{F}}_t((j,m))$, Equation (7) follows.

It remains to show that the ET can indeed compute Equation (8). To this end, we will require a single transformer head in each layer. Specifically, we want this head to compute

$$h_1(\boldsymbol{X}^{(t-1)})_{jm} = \frac{\beta}{n}\sum_{l=1}^{n} \boldsymbol{T}^{(t-1)}_{jl} \cdot \Big(\boldsymbol{T}^{(t-1)}_{lm}\Big)^{n^2}. \tag{9}$$

Now, recall the definition of the Edge Transformer head at tuple $(j,m)$ as

$$h_1(\boldsymbol{X}^{(t-1)})_{jm} := \sum_{l=1}^{n} \alpha_{jlm} \boldsymbol{V}^{(t-1)}_{jlm},$$

where

$$\alpha_{jlm} := \underset{l \in [n]}{\text{softmax}}\Big(\frac{1}{\sqrt{d_k}} \boldsymbol{X}^{(t-1)}_{jl} \boldsymbol{W}^Q (\boldsymbol{X}^{(t-1)}_{lm} \boldsymbol{W}^K)^T\Big)$$

with

$$\boldsymbol{V}^{(t-1)}_{jlm} := \boldsymbol{X}^{(t-1)}_{jl}\begin{bmatrix}\boldsymbol{W}^{V_1}_1 \\ \boldsymbol{W}^{V_1}_2\end{bmatrix} \odot \boldsymbol{X}^{(t-1)}_{lm}\begin{bmatrix}\boldsymbol{W}^{V_2}_1 \\ \boldsymbol{W}^{V_2}_2\end{bmatrix}$$

and by the induction hypothesis above,

$$\boldsymbol{X}_{jl}^{(t-1)} = \left[\boldsymbol{T}_{jl}^{(t-1)} \quad \left(\boldsymbol{T}_{jl}^{(t-1)}\right)^{n^2}\right]$$

$$\boldsymbol{X}_{lm}^{(t-1)} = \left[\boldsymbol{T}_{lm}^{(t-1)} \quad \left(\boldsymbol{T}_{lm}^{(t-1)}\right)^{n^2}\right],$$

where we expanded sub-matrices. Specifically, $\boldsymbol{W}_1^{V_1}, \boldsymbol{W}_1^{V_2}, \boldsymbol{W}_2^{V_1}, \boldsymbol{W}_2^{V_2} \in \mathbb{R}^{\frac{d}{2} \times d}$. We then set

$$\boldsymbol{W}^Q = \boldsymbol{W}^K = \boldsymbol{0}$$
$$\boldsymbol{W}_1^{V_1} = [\beta \boldsymbol{I} \quad \boldsymbol{0}]$$
$$\boldsymbol{W}_2^{V_1} = [\boldsymbol{0} \quad \boldsymbol{0}]$$
$$\boldsymbol{W}_1^{V_2} = [\boldsymbol{0} \quad \boldsymbol{I}]$$
$$\boldsymbol{W}_2^{V_2} = [\boldsymbol{0} \quad \boldsymbol{0}].$$

Here, $\boldsymbol{W}^Q$ and $\boldsymbol{W}^K$ are set to zero to obtain uniform attention scores. Note that then for all $j, l, k$, $\alpha_{jlm} = \frac{1}{n}$, due to normalization over $l$, and we end up with Equation (9) as

$$h_1(\boldsymbol{X}^{(t-1)})_{jm} = \frac{1}{n} \sum_{l=1}^{n} \boldsymbol{V}_{jlm}^{(t-1)}$$

where

$$\boldsymbol{V}_{jlm}^{(t-1)} = \left[\boldsymbol{T}_{jl}^{(t-1)} \cdot \beta \boldsymbol{I} + \left(\boldsymbol{T}_{jl}^{(t-1)}\right)^{n^2} \cdot \boldsymbol{0} \quad \boldsymbol{0}\right] \odot \left[\boldsymbol{T}_{lm}^{(t-1)} \cdot \boldsymbol{0} + \left(\boldsymbol{T}_{lm}^{(t-1)}\right)^{n^2} \cdot \boldsymbol{I} \quad \boldsymbol{0}\right]$$

$$= \beta \cdot \left[\boldsymbol{T}_{jl}^{(t-1)} \cdot \left(\boldsymbol{T}_{lm}^{(t-1)}\right)^{n^2} \quad \boldsymbol{0}\right].$$

We now conclude our proof as follows. Recall that the Edge Transformer layer computes the final representation $\boldsymbol{X}^{(t)}$ as

$$\boldsymbol{X}_{jm}^{(t)} = \mathsf{FFN}\left(\boldsymbol{X}_{jm}^{(t-1)} + h_1(\boldsymbol{X}^{(t-1)})_{jm} \boldsymbol{W}^O\right)$$

$$= \mathsf{FFN}\left(\left[\boldsymbol{T}_{jm}^{(t-1)} \quad \left(\boldsymbol{T}_{jm}^{(t-1)}\right)^{n^2}\right] + \frac{\beta}{n} \sum_{l=1}^{n} \left[\boldsymbol{T}_{jl}^{(t-1)} \cdot \boldsymbol{T}_{lm}^{(t-1)} \quad \boldsymbol{0}\right] \boldsymbol{W}^O\right)$$

$$\underset{\boldsymbol{W}^O := \boldsymbol{I}}{=} \mathsf{FFN}\left(\left[\boldsymbol{T}_{jm}^{(t-1)} \quad \left(\boldsymbol{T}_{jm}^{(t-1)}\right)^{n^2}\right] + \left[\frac{\beta}{n} \sum_{l=1}^{n} \boldsymbol{T}_{jl}^{(t-1)} \cdot \boldsymbol{T}_{lm}^{(t-1)} \quad \boldsymbol{0}\right]\right)$$

$$= \mathsf{FFN}\left(\left[\boldsymbol{T}_{jm}^{(t-1)} + \frac{\beta}{n} \sum_{l=1}^{n} \boldsymbol{T}_{jl}^{(t-1)} \cdot \boldsymbol{T}_{lm}^{(t-1)} \quad \left(\boldsymbol{T}_{jm}^{(t-1)}\right)^{n^2}\right]\right)$$

$$\underset{Eq.8}{=} \mathsf{FFN}\left(\left[\boldsymbol{T}_{jm}^{(t)} \quad \left(\boldsymbol{T}_{jm}^{(t-1)}\right)^{n^2}\right]\right)$$

for some FFN. Note that the above derivation only modifies the terms inside the parentheses and is thus independent of the choice of FFN. We have thus shown that the ET can compute Equation (8).

To complete the induction, let $f : \mathbb{R}^2 \to \mathbb{R}^2$ be such that

$$f\left(\left[\boldsymbol{T}_{jm}^{(t)} \quad \left(\boldsymbol{T}_{jm}^{(t-1)}\right)^{n^2}\right]\right) = \left[\boldsymbol{T}_{jm}^{(t)} \quad \left(\boldsymbol{T}_{jm}^{(t)}\right)^{n^2}\right].$$

Since our domain is compact, $f$ is continuous, and hence we can choose FFN to approximate $f$ arbitrarily close. This completes the proof. $\square$

Next, we show the other direction of Theorem 1 under mild and reasonable assumptions. First, we say that a recoloring function, that maps structures over positive integers into positive integers, is *(effectively) invertible* if its inverse is computable. All coloring functions used in practice (e.g., hash-based functions, those based on pairing functions, etc) are invertible. Second, the layer normalization operation is a proper function if it uses statistics collected only during training mode, and not during evaluation mode.

**Proposition 8.** *Let* recolor *be an invertible function, and let us consider the 2-FWL coloring algorithm using* recolor. *Then, for all parametrizations of the ET with proper layer normalization, for all node-labeled graphs $G = (V(G), E(G), \ell)$, and for all $t \geq 0$:*

$$C_t^{2,F}(\boldsymbol{v}) = C_t^{2,F}(\boldsymbol{w}) \Longrightarrow \boldsymbol{X}^{(t)}(\boldsymbol{v}) = \boldsymbol{X}^{(t)}(\boldsymbol{w}),$$

*for all pairs of 2-tuples $\boldsymbol{v}$ and $\boldsymbol{w}$ in $V(G)^2$.*

*Proof.* We first claim that there is a computable function $Z : \mathbb{N}^* \times \mathbb{N} \to \mathbb{R}^p$, where $\mathbb{N}^* = \{0\} \cup \mathbb{N}$, such that $\boldsymbol{X}^{(t)}(\boldsymbol{v}) = Z(t, C_t^{2,F}(\boldsymbol{v}))$ for all $\boldsymbol{v} \in V(G)^2$, independent of the graph $G$ and its order. The proof of the claim is by induction on $t$. For $t = 0$, by definition, $C_0^{2,F}(\boldsymbol{v})$ identifies the atomic type $\text{atp}_2(\boldsymbol{v})$ which defines $\boldsymbol{X}^{(0)}(\boldsymbol{v})$ (since the atomic type tells if $v$ is an edge in $G$, and the labels of the vertices in $\boldsymbol{v}$).

For $t > 0$ and $\boldsymbol{v} = (i, j)$, the function $Z(t, C_t^{2,F}(\boldsymbol{v}))$ proceeds as follows. First, it uses the invertibility of recolor to obtain the pair

$$\left( C_{t-1}^{2,F}(i,j), \; \{\!\!\{ \left( C_{t-1}^{2,F}(i,l), C_{t-1}^{2,F}(l,j) \right) \mid l \in V(G) \}\!\!\} \right).$$

Then, by inductive hypothesis using the function $Z(t-1, \cdot)$, it obtains the pair

$$\left( \boldsymbol{X}^{(t-1)}(i,j), \; \{\!\!\{ \left( \boldsymbol{X}^{(t-1)}(i,l), \boldsymbol{X}^{(t-1)}(l,j) \right) \mid l \in V(G) \}\!\!\} \right).$$

Finally, it computes

$$\boldsymbol{X}^{(t)}(i,j) = \mathsf{FFN}\left( \boldsymbol{X}^{(t-1)}(i,j) + \mathsf{TriAttention}\left( \mathsf{LN}\left( \boldsymbol{X}^{(t-1)}(i,j) \right) \right) \right)$$

under the assumption that the layer normalization is a proper function. The statement of the proposition then follows directly from the claim since

$$\boldsymbol{X}^{(t)}(\boldsymbol{v}) = Z(t, C_t^{2,F}(\boldsymbol{v})) = Z(t, C_t^{2,F}(\boldsymbol{w})) = \boldsymbol{X}^{(t)}(\boldsymbol{w}).$$

$\square$

Note that unlike the result in Proposition 7, the above result is uniform, in that the concrete choice of recolor and the function $Z$ does not depend on the graph size $n$. Finally, Theorem 1 follows from Proposition 7 and Proposition 8.

