# OpenReview forum: "Towards Principled Graph Transformers"
_NeurIPS.cc/2024/Conference — NeurIPS 2024 poster_

### Official Review · Reviewer_ckA7 · 2024-07-09

**Soundness:** 2
**Presentation:** 2
**Contribution:** 2
**Rating:** 1
**Confidence:** 5

**Summary:**

This paper introduces a novel architecture called Edge Transformer (ET) for graph learning tasks. The ET leverages global attention on node pairs instead of individual nodes and demonstrates strong empirical performance without relying on positional or structural encodings. The authors show that ET has the expressive power of 3-WL and achieves competitive results on algorithmic reasoning and molecular regression tasks.

**Strengths:**

1、ET bridges the gap between theoretical expressivity and practical performance.
2、The paper provides comprehensive empirical evidence demonstrating that the Edge Transformer outperforms existing theoretically motivated models and competes well with state-of-the-art models in various graph learning tasks.

**Weaknesses:**

1、The total number of parameters for the model is not provided in the parameter table.
2、While ET is more efficient than some other high-order transformers, it is still significantly inferior to most graph transformers with O(n^2). This limits the applicability of ET.
3、The author's contribution is confined to applying the ET model to the graph, without proposing their own method.

**Questions:**

1、Can you provide possible ways to reduce the complexity of ET?
2、Can you provide the total number of parameters for ET on each dataset? See Weaknesses 1.
3、Why has the performance changed so much compared to the submitted version of ICML in 2024? Especially on the Searching dataset.
4、What do Average and All algorithms represent respectively?

**Limitations:**

See Weaknesses.

---

> ### Author Rebuttal · Authors · 2024-08-06
>
> We thank the reviewer for their effort and are happy to address the concerns raised by the reviewer. First, the reviewer states
>
> > The total number of parameters for the model is not provided in the parameter table
> >
>
> We provide the parameters for each model in the general response above. As can be observed, on the molecular regression tasks, the ET is well within the 500K parameter budget.
>
> Next, the reviewers says
>
> > While ET is more efficient than some other high-order transformers, it is still significantly inferior to most graph transformers with O(n^2). This limits the applicability of ET
> >
>
> See our general response for a discussion of the practical usefulness of the ET. In summary, we argue that there exists still plenty of application and interest for a model with high runtime complexity, particularly, if this complexity can be translated into strong predictive performance.
>
> Finally, the reviewer says
>
> > The author's contribution is confined to applying the ET model to the graph, without proposing their own method
> >
>
> See our general response for a discussion on the novelty and contribution of our work. In summary, we argue that the novelty of our work does not come from proposing a new method. Rather, the main contribution of our work is in providing a connection between the ET, systematic generalization and graph learning.
>
> ### Questions
>
> Further, we are happy to address the question of the reviewer below. First, the reviewer asks
>
> > Can you provide possible ways to reduce the complexity of ET?
> >
>
> As our work is partly motivated by demonstrating that high expressivity can also result in strong empirical performance, we do not aim to reduce the theoretical complexity of the ET in this work. At the same time, we show that in practice, the high runtime and memory cost can be reduced by leveraging a parallelizable implementation, together with low-level GPU optimizations; see the limitation section in our paper. More generally, while the ET may not be applicable to large graphs yet, we regard our work as a first step at a theoretical understanding of equivariant graph transformers with high expressive power that also perform strongly in practice.
>
> Next, the reviewer asks
>
> > Why has the performance changed so much compared to the submitted version of ICML in 2024? Especially on the Searching dataset.
> >
>
> In a previous version, we had not included the graph features that are available in CLRS, and used by baseline models, into the ET. These features proved particularly useful on the Searching datasets, explaining the improved performance.
>
> Finally, the reviewer asks
>
> > What do Average and All algorithms represent respectively?
> >
>
> In “Average”, we average performance across the six algorithm classes, whereas in “All algorithms” we average performance over all algorithms.
>
> We want to ask the reviewer to increase their score if they are satisfied with our response. We are happy to answer any remaining questions.

---

> > ### Comment · Reviewer_ckA7 · 2024-08-12
> >
> > Considering that the performance of this version achieves about 30% improvement than that of ICML version on Searching dataset, while it decreases on Strings and Geometry datasets, I believe the emprical results are not convincing. Moreover, I think the response is not reasonable.

---

### Official Review · Reviewer_Ff1y · 2024-07-10

**Soundness:** 3
**Presentation:** 3
**Contribution:** 2
**Rating:** 6
**Confidence:** 4

**Summary:**

In this paper, the authors show that Edge Transformer, a global attention model operating on node pairs, has 3-WL expressive power when provided with the right tokenization. Experiments results also show that the Edge Transformer has competitive performance on  molecular regression tasks and algorithmic reasoning.

**Strengths:**

1.  This paper proposes a concrete implementation of the Edge Transformer, and proves that it has an expressive power of 3-WL.
2.  The proposed ET has superior empirical performance on molecular regression and neural algorithmic reasoning tasks.

**Weaknesses:**

1. My major concern is the real usefulness of ET, since the high runtime and memory complexity may offset the importance of ET, though the authors discuss it in Limitations. Also, it is not clear what is the parameter count, which raises the question that the better performance of ET may be mainly from large number of parameters.
2. It seems that the ET in this paper is mainly from existing work, and the difference needs to be explained.

**Questions:**

1. The ET in this paper is a variant of existing ones, due to the specific tokenization. But it is not quite clear to me what are the main differences of the tokenization compared with existing ones, and why this tokenization can help the 3-WL. It would be good if the authors provide more details.
2. In Table 1, ET with RRWP has a little bit lower performance on QW9. Can the authors elaborate on this? Since generally, transformers with positional encodings will lead to improved performance.
3. The authors emphasize that the designed ET can achieve superior performance even without positional encodings. But maybe as an open question, if incorporated with PEs, will the theoretical results change?

[1] Comparing Graph Transformers via Positional Encodings, ICML 2024

**Limitations:**

Scalability limitations are discussed.

---

> ### Author Rebuttal · Authors · 2024-08-06
>
> We thank the reviewer for their effort and are happy to address the concerns raised by the reviewer. First, the reviewer states
>
> > My major concern is the real usefulness of ET, since the high runtime and memory complexity may offset the importance of ET
> >
>
> See our general response for discussion of the practical usefulness of the ET. In summary, we argue that there exists still plenty of application and interest for a model with high runtime complexity, particularly, if this complexity can be translated into strong predictive performance.
>
> Next, the reviewer says
>
> > Also, it is not clear what is the parameter count, which raises the question that the better performance of ET may be mainly from large number of parameters
> >
>
> We provide the parameter counts in the general response. We want to highlight that on ZINC and Alchemy, we adhere to the 500K parameter budget and that the ET’s strong performance cannot be explained by a comparatively large number of parameters.
>
> Finally, the reviewer says
>
> > It seems that the ET in this paper is mainly from existing work, and the difference needs to be explained
> >
>
> See our general response for a discussion on the novelty and contribution of our work. In summary, we argue that the novelty of our work does not come from proposing a new method. Rather, the main contribution of our work is in providing a connection between the ET, systematic generalization and graph learning. To still answer the question about the difference between the ET in our work and the original ET, note that on the architectural side, we provide a sufficient tokenization and readout for the ET to simulate 2-FWL. In what follows, we provide a detailed explanation in response to the reviewer’s first question, which also touches on this concern.
>
> ### Questions
>
> The reviewer asks
>
> > But it is not quite clear to me what are the main differences of the tokenization compared with existing ones, and why this tokenization can help the 3-WL
> >
>
> We are happy to provide further details on the proposed tokenization for the ET. For the tokenization to be sufficient for the 2-FWL, the initial embeddings of each node pair need to injectively encode the initial colors of each node, as well as an indicator of whether the nodes are connected by an edge. Such a tokenization is already vaguely described in the original paper (https://arxiv.org/abs/2112.00578). In this work, we merely formalize this tokenization such that it can be used in our formal proofs. Other than the tokenization and readout, we do not modify the ET at all.
>
> Next, the reviewer asks
>
> > In Table 1, ET with RRWP has a little bit lower performance on QW9. Can the authors elaborate on this? Since generally, transformers with positional encodings will lead to improved performance
> >
>
> We suspect that adding positional encodings (PEs) only translates to improved empirical performance if the PEs provide additional features either useful for expressivity or directly for prediction. However, since the ET already has a high base expressivity, on this particular task, the additional RRWP encodings may not be useful for either distinguishing graphs or directly for predicting the targets and instead lead to slight overfitting. A takeaway from this finding may be that the base expressivity of the ET is often enough, as the RRWP encodings do not prove substantial for the good performance of the ET. On the other hand, models such as the SOTA graph transformer GRIT rely on PEs for good performance; see https://arxiv.org/abs/2305.17589, Table 5.
>
> Finally, the reviewer asks
>
> > But maybe as an open question, if incorporated with PEs, will the theoretical results change?
> >
>
> We agree with the reviewer that this is an interesting question for future work. In particular, if one were able to show that RRWP encodings can distinguish graphs indistinguishable by the 3-WL, one would obtain a theoretical guarantee that ET+RRWP leads to increased expressivity. At the same time, we regard it as a strength of the ET that the model does not *rely* on PEs but can still successfully leverage them, if provided and informative on the given task; see e.g., our results on ZINC 12K.
>
> We want to ask the reviewer to increase their score if they are satisfied with our response. We are happy to answer any remaining questions.

---

> > ### Comment · Reviewer_Ff1y · 2024-08-12
> >
> > Thank the authors for clarifications and explaining the main contributions. Though this paper does not propose a new model, it is a solid paper. I am happy to increase my rating.

---

### Official Review · Reviewer_szHE · 2024-07-11

**Soundness:** 2
**Presentation:** 4
**Contribution:** 2
**Rating:** 5
**Confidence:** 4

**Summary:**

This paper applies Edge Transformer (ET) to the field of graph learning. It is proved that ET has 3-WL expressivity with cubic complexity, which is more efficient than existing 3-WL models. Experiments on BREC benchmark clearly demonstrate its expressive power. Results on the other three molecular datasets and CLRS benchmark are also provided.

**Strengths:**

- It proves ET has 3-WL expressive power and demonstrates excellent results on BREC benchmark. Even though it has $O(n^3)$ complexity, it is still more efficient than other 3-WL models.
- It shows good performance on real-world graph learning tasks.
- The paper is well-written and easy to follow.

**Weaknesses:**

- My major concern is about empirical evaluation.
    - While I appreciate ET's excellent theoretical expressive power, I find it is needed to include more benchmark datasets in graph learning (e.g., OGB, moleculeNet, GNN benchmark, Long Rang Graph Benchmark, Zinc-full, etc) to comprehensively demonstrate its practical predictive power compared with state-of-the-art (SOTA) methods, e.g., those graph transformers Grit, Exphormer, GraphGPS, etc.
    - Besides datasets from algorithmic reasoning, currently, in my opinion, ET is only evaluated on two widely-used graph benchmark datasets with performance reported from very recent SOTA methods, i.e., Zinc-12k and Alchemy-12k, making it hard to assess ET's practical performance. Although QM9 is also a popular dataset, the adopted setting is less common. Meanwhile, Zinc-12k is just a small subset of Zinc-full, and it has recently been shown that being good at Zinc-12k does not necessarily lead to the best performance on Zinc-full [1].
    - It would be interesting to see its long-range modeling capability on the LRGB benchmark as it's a transformer.
    - In terms of the evaluation of expressive power, it would be better to include SOTA methods such as those subgraph-GNNs, potentially also discussing their complexity compared to ET.


[1] Wang, X., Li, P., & Zhang, M. (2024). Graph as Point Set. ICML 2024.

**Questions:**

- How does the setting of QM9 used in this paper differ from the setting that is commonly used in equivariant GNNs (e.g., EGNN, DimeNet, etc)? It would be better if this description could be put in the paper, making it self-contained.
- Are there any particular benefits to having an expressive model without using PE or SE? I consider the major benefit would be on the computation side as computing PE can be $O(n^2)$, but ET has already had $O(n^3)$ complexity. So, for the claim in the paper, is it just because it would be more challenging to analyze expressive power with PE/SE, therefore it would be easier to understand a model's expressive power without using them?
    - I feel like depending on how PE is used, it is still possible to analyze the expressive power, e.g.,  via graph substructure counting. Can the authors discuss the works such as graph as point set and those subgraph-GNNs? Because they may also use PE, while proving their expressive power via graph substructure counting.

**Limitations:**

See above.

---

> ### Author Rebuttal · Authors · 2024-08-07
>
> We thank the reviewer for their effort and are happy to address the concerns raised by the reviewer.
>
> Regarding evaluation, the reviewer says that
>
> > I find it is needed to include more benchmark datasets in graph learning […] to comprehensively demonstrate its practical predictive power compared with state-of-the-art (SOTA) methods, e.g., those graph transformers Grit, Exphormer, GraphGPS
> >
>
> as well as
>
> > Besides datasets from algorithmic reasoning, currently, in my opinion, ET is only evaluated on two widely-used graph benchmark datasets
> >
>
> and
>
> > It would be interesting to see its long-range modeling capability on the LRGB benchmark as it's a transformer
> >
>
> While we certainly evaluate the ET only on a subset of all possible graph learning benchmarks we argue that our results already provide a comprehensive view on the predictive performance of the ET. In particular, we evaluate the ET on three different types of benchmarks:
>
> - Molecular property prediction (ZINC, Alchemy, QM9)
> - Algorithmic reasoning (CLRS)
> - Graph isomorphism testing (BREC)
>
> Most notably, the CLRS benchmark contains a total of 30 different datasets and includes a variety of graph-, node- and edge-level tasks. In addition, the CLRS benchmark inherently evaluates models on size generalization, a problem recently highlighted as an important challenge in graph learning; see https://arxiv.org/abs/2402.02287, Section 3.1, Challenge III.4. Hence, we argue that CLRS is a high-quality benchmark and note that it is also widely adopted; see e.g., https://arxiv.org/abs/2209.11142, https://arxiv.org/abs/2403.04929, https://arxiv.org/abs/2402.01107, https://arxiv.org/abs/2404.03441, https://arxiv.org/abs/2312.05611, https://arxiv.org/abs/2210.05062.
>
> Finally, note that our concrete selection of tasks is partly also to validate our theoretical findings and not only to demonstrate SOTA performance on the most popular datasets; see also our remark in the general response on the intended scope of our work, in particular, regarding an insightful theoretical and empirical investigation of the ET.
>
> The above being said, we worked hard during the rebuttal to provide encouraging preliminary results on two additional datasets, see below.
>
> ### PCQM4Mv2
>
> We evaluate on one random seed, as is common on this benchmark.
>
> | Model | Num. parameters | Validation MAE |
> | --- | --- | --- |
> | GraphGPS-small | 6M | 0.0938 |
> | GRIT | 17M | 0.0859 |
> | TokenGT | 50M | 0.0910 |
> | ET | 6M | 0.0911 |
> | ET | 17M | 0.0883 |
>
> ### ZINC-full
>
> We evaluate on 2 random seeds. The ET has 283,969 parameters.
>
> | Model | Test MAE |
> | --- | --- |
> | SignNet | 0.024 $\pm$ 0.003 |
> | Graphormer | 0.052 $\pm$ 0.005 |
> | Graphormer-GD | 0.025 $\pm$ 0.004 |
> | GRIT | 0.023 $\pm$ 0.001 |
> | PST | 0.018 $\pm$ 0.001 |
> | ET+RRWP | 0.028 $\pm$ 0.002 |
>
> Notably, on PCQM4Mv2 our 6M model outperforms the GraphGPS model at the same parameter budget while also being on par with transformer models such as TokenGT which has almost 5x as many parameters. On ZINC-full, we observe that the results of the ET are competitive with the best models, albeit not SOTA results.
>
> We note that achieving SOTA results on new datasets typically involves extensive hyper-parameter tuning. Due to the limited time during the rebuttal we were only able to test a few configurations and also at only a fraction of the number of epochs (1000 instead of 2000 on ZINC-full and 40 instead of 200 on PCQM4Mv2) of the baseline models. Nonetheless, our results indicate that with more time for properly tuning hyper-parameters and running the same number of epochs, the ET is a viable contender for SOTA results on both datasets.
>
> Finally, the reviewer says
>
> > In terms of the evaluation of expressive power, it would be better to include SOTA methods such as those subgraph-GNNs, potentially also discussing their complexity compared to ET
> >
>
> We refer the reviewer to Wang and Zhang 2023 (https://arxiv.org/abs/2304.07702), Table 2. Here, the authors have already evaluated a variety of subgraph GNNs on BREC. Most notably, the ET beats 7 out of 9 models on this benchmark.
>
> In what follows, we address the question of the reviewer. First, the reviewer asks
>
> > How does the setting of QM9 used in this paper differ from the setting that is commonly used in equivariant GNNs
> >
>
> In this work, for QM9, we adopt the setting used for SpeqNets (https://arxiv.org/abs/2203.13913), where tasks are learnt jointly and the resulting MAE is the average across all tasks, where we note that all targets are normalized by subtracting the mean and dividing by the standard deviation. We follow this setting to enable a comparison to the higher-order WL SpeqNets. We will add this description to the paper and plan to include results on QM9 in the more widely adopted single-task setting in the future.
>
> Next, the reviewer asks
>
> > Are there any particular benefits to having an expressive model without using PE or SE? […] because it would be more challenging to analyze expressive power with PE/SE, therefore it would be easier to understand a model's expressive power without using them?
> >
>
> and mentions that
>
> > I feel like depending on how PE is used, it is still possible to analyze the expressive power, e.g., via graph substructure counting
> >
>
> While it is true that the expressive power of PEs can be understood, in general, we argue that knowing which PEs to add is typically highly task-dependent; see e.g., the study in GraphGPS (https://arxiv.org/abs/2205.12454), as PEs are typically based on graph heuristics. This also holds for subgraph GNNs, where assumptions have to be made about the type of substructures that are relevant for downstream tasks. In contrast, the ET without any additional PEs already performs strongly, across tasks in molecular regression, algorithmic reasoning and graph isomorphism testing.
>
> We want to ask the reviewer to increase their score if they are satisfied with our response. We are happy to answer any remaining questions.

---

> > ### Comment · Reviewer_szHE · 2024-08-12
> >
> > I have read the authors' response and other reviews and intend to keep the original rating.

---

### Official Review · Reviewer_YYTZ · 2024-07-12

**Soundness:** 3
**Presentation:** 3
**Contribution:** 3
**Rating:** 7
**Confidence:** 4

**Summary:**

This work proposes an Edge Transformer (ET), a global attention model operating on node pairs instead of nodes. Authors theoretically demonstrate that ET has 3-WL expressive power with the proper tokenization. Experimental results demonstrate that the proposed model outperforms other theoretically aligned models in terms of predictive performance. Additionally, ET competes with state-of-the-art models in algorithmic reasoning and molecular regression tasks without relying on positional or structural encodings.

**Strengths:**

1. The paper is well-written. All technical steps are easy to follow.
2. The proposed triangular attention is interesting and novel.
3. ET achieves impressive results for both molecular regression and algorithmic reasoning tasks.

**Weaknesses:**

1. Previous work [1] has demonstrated that graph transformers (GTs) with proper positional encodings (PEs) can be more powerful than any WL test. Therefore, it is unclear why the authors aim to develop a model with 3-WL expressive power, which cannot be more expressive than prior GTs (e.g., [1]). Notably, given the cubic complexity of the proposed ET, computing PEs is no longer a scalability bottleneck. Consequently, if ET empirically outperforms GT+PE, it essentially implies that the WL hierarchy is not a meaningful metric in practice, undermining the purpose of building a 3-WL expressive model.
2. While the authors mention how to apply ET for node-level tasks, they provide no corresponding results. Given the cubic complexity of ET, it is likely not scalable to most graphs for node-level tasks.
3. Some relevant GT models haven't been discussed or compared (e.g., [1-3]).

[1] Kreuzer et al., "Rethinking Graph Transformers with Spectral Attention", NeurIPS'21. \
[2] Zhu et al., "On Structural Expressive Power of Graph Transformers", KDD'23. \
[3] Deng et al., "Polynormer: Polynomial-Expressive Graph Transformer in Linear Time", ICLR'24.

**Questions:**

N/A

**Limitations:**

Please refer to the Weaknesses section for details.

---

> ### Author Rebuttal · Authors · 2024-08-06
>
> We thank the reviewer for their effort and are happy to address the concerns raised by the reviewer. First, the reviewer says
>
> > Previous work [1] has demonstrated that graph transformers (GTs) with proper positional encodings (PEs) can be more powerful than any WL test
> >
>
> The work in [1] presents a universality result for transformers with Laplacian encodings. However, the transformer in [1] is not equivariant to node permutations, since the Laplacian encodings are not invariant to sign- and basis changes in the underlying eigenvalue decomposition. While a universal model is more expressive than any WL test, it is questionable whether the transformer in [1] is able to learn permutation equivariance from limited data. Theoretically, we expect worse generalization from non-equivariant models on graph tasks; see e.g., Petrache and Trivedi [2]. In fact, the lack of permutation equivariance could well explain the worse empirical performance of the model in [1] compared to the ET, e.g., on ZINC 12K. In summary, our theoretical results should be viewed under the assumption of permutation equivariance, where 3-WL is considered to have very strong expressive power.
>
> Next, the reviewer says
>
> > While the authors mention how to apply ET for node-level tasks, they provide no corresponding results
> >
>
> We refer the reviewer to the experiments on the CLRS benchmark. There, many tasks involve node- and edge-level predictions. The reviewer also mentions that
>
> > Given the cubic complexity of ET, it is likely not scalable to most graphs for node-level tasks.
> >
>
> We agree that the ET currently cannot be scaled to thousands of nodes and that most large-scale transductive node tasks are out of scope for the ET. Nonetheless, our results on CLRS indicate that the ET can be effectively used for node-level tasks.
>
> Finally, the reviewer mentions that
>
> > Some relevant GT models haven't been discussed or compared (e.g., [1-3])
> >
>
> We are happy to include those models into our discussion. In particular, we will add a paragraph discerning expressivity results for permutation equivariant models as compared to non-equivariant but universal models such as the one in [1].
>
> We want to ask the reviewer to increase their score if they are satisfied with our response. We are happy to answer any remaining questions.
>
> ### References
>
> [1] Kreuzer et al., **Rethinking Graph Transformers with Spectral Attention,** 2021 https://arxiv.org/abs/2106.03893
>
> [2] Petrache and Trivedi, **Approximation-Generalization Trade-offs under (Approximate) Group Equivariance,** 2023 https://arxiv.org/abs/2305.17592

---

> ### Comment · Reviewer_YYTZ · 2024-08-10
> **Follow-up**
>
> Thanks for authors' detailed response. By node-level tasks, I’m actually referring to some larger graphs for node classification (e.g., citation/biological networks) rather than neural algorithmic reasoning tasks. I’m interested in seeing how well the proposed approach scales with larger graphs.
>
> >We want to ask the reviewer to increase their score if they are satisfied with our response.
>
> I'm really not a big fan of explicitly asking reviewers to change their scores in this way. Authors could say "reconsidering their score" instead.
>
> That said, I do appreciate the major contributions of this work and increase my score to 7.

---

### Author Rebuttal · Authors · 2024-08-06

We thank all reviewers for their time and effort and the valuable feedback they provided for our work. Here, we address two common concerns. In addition, we provide parameter counts for each dataset.

### On the novelty and contribution of our work

Here, we address the concern that our work is not novel because we do not propose a new architecture.

Indeed, we adopt the existing ET architecture, with almost no modifications. At the same time, we want to point out that novelty does not solely originate from proposing novel architectures. Concretely, we regard the main contribution of our work as establishing the connection between the ET, its systematic generalization capabilities and graph learning (Sections 3 and 4). In particular, we provide two highly non-trivial proofs that show that the ET has exactly 3-WL expressive power. Our empirical study (Section 5) is aimed at both validating the theoretical findings, as well as demonstrating that the ET can effectively translate its expressive power into strong empirical performance. As such, our work should be viewed as a theoretical and empirical investigation rather than proposing a novel method.

### On the practical usefulness of the ET

Here, we address the concern of the practical usefulness of the ET and the results in our paper, given its cubic runtime and memory complexity.

As already addressed in our limitations section, the ET has indeed a high runtime and memory complexity. At the same time, we want to highlight that the ET shows very strong performance on smaller graphs, in particular, on molecular graphs. As such, we believe the ET to be very useful in the context of molecular representation learning. Here, we want to highlight TGT (https://arxiv.org/abs/2402.04538) and AlphaFold 3 (https://www.nature.com/articles/s41586-024-07487-w), two recent works that use models with a triangular attention mechanism for learning molecular representations. In particular, both works compute attention scores over triplets of nodes. Although the attention mechanism in these works differs from the one in the ET, their strong performance demonstrates nonetheless the usefulness of studying triangular-style attention and its impact in molecular representation learning. Most notably, the authors of TGT refer to https://arxiv.org/abs/2302.05743 to motivate their work, arguing that

> 3rd-order interactions are crucial for geometric understanding, yet ≥4th-order interactions add little/no benefit at much higher computational cost
>

In addition, the ET performs particularly well on CLRS. We refer the reviewer to the CLRS paper (https://arxiv.org/abs/2205.15659) as well to https://arxiv.org/abs/2105.02761, for an in-depth discussion of the potential use-cases of algorithmic reasoning in neural networks.

### Number of parameters

Here, we report the number of parameters of the ET without positional encodings.

| Dataset | Num. parameters |
| --- | --- |
| ZINC 12K | 280,897 |
| Alchemy 10K | 295,356 |
| QM9 | 257,916 |
| CLRS | 1,371,264 |
| BREC | 58,112 |

Here, we report the number of parameters of ET+RRWP.

| Dataset | Num. parameters |
| --- | --- |
| ZINC 12K | 283,969 |
| Alchemy 10K | 298,428 |
| QM9 | 260,988 |

---

### Decision · Program_Chairs · 2024-09-25

**Decision:**

Accept (poster)

**Comment:**

This paper analyzes the expressive power of Edge Transformer (graph transformer operating on node pairs) and shows that it has equivalent expressive power as 3-WL without using PEs. The paper receives 1, 7, 5, 6 scores after rebuttal, with reviewer ckA7 decreasing their score from 3 to 1. I carefully checked the reviews and rebuttals and deemed that the review content does not justify the score, and the authors have successfully responded to ckA7's concerns. Further, reviewer ckA7 did not participate into the discussion. Therefore, their score is downweighted and a final acceptance recommendation is made.

Overall, I encourage the authors to include their new results in the rebuttal, propertly address all the reviewers' concerns, and work further on addressing the computational complexity of ET to make it practically more useful (on large graphs).